# Insights into real-time chemical processes in a calcium sensor protein-directed dynamic library

Andrea Canal-Martín[1,2], Javier Sastre [1], María José Sánchez-Barrena[3], Angeles Canales[2], Sara Baldominos[1], Naiara Pascual[1], Loreto Martínez-González[1], Dolores Molero[4], Mª Encarnación Fernández-Valle[4], Elena Sáez[4], Patricia Blanco-Gabella[3], Elena Gómez-Rubio[1], Sonsoles Martín-Santamaría[1], Almudena Sáiz[5], Alicia Mansilla[5], F. Javier Cañada [1], Jesús Jiménez-Barbero[6], Ana Martínez [1] & Ruth Pérez-Fernández[1]

Dynamic combinatorial chemistry (DCC) has proven its potential in drug discovery speeding the identification of modulators of biological targets. However, the exchange chemistries typically take place under specific reaction conditions, with limited tools capable of operating under physiological parameters. Here we report a catalyzed protein-directed DCC working at low temperatures that allows the calcium sensor NCS-1 to find the best ligands in situ. Ultrafast NMR identifies the reaction intermediates of the acylhydrazone exchange, tracing the molecular assemblies and getting a real-time insight into the essence of DCC processes at physiological pH. Additionally, NMR, X-ray crystallography and computational methods are employed to elucidate structural and mechanistic aspects of the molecular recognition event. The DCC approach leads us to the identification of a compound stabilizing the NCS-1/Ric8a complex and whose therapeutic potential is proven in a *Drosophila* model of disease with synaptic alterations.

[1] Structural and Chemical Biology Department, Centro de Investigaciones Biológicas, CIB-CSIC, Madrid 28040, Spain. [2] Organic Chemistry Department, Universidad Complutense de Madrid, Madrid 28040, Spain. [3] Department of Crystallography and Structural Biology, Instituto de Química Física Rocasolano, IQFR-CSIC, Madrid 28006, Spain. [4] CAI de RMN, Universidad Complutense de Madrid, 28040 Madrid, Spain. [5] Instituto Ramón y Cajal de Investigación Sanitaria. Ctra. Colmenar Viejo, km. 9100, 28034 Madrid, Spain. [6] Molecular recognition and host-pathogen interactions, CIC bioGUNE, Derio 48160 Bizkaia, Spain. Correspondence and requests for materials should be addressed to M.J.S.-B. (email: xmjose@iqfr.csic.es) or to A.M. (email: alicia.mansilla@salud.madrid.org) or to R.P.-F. (email: ruth.perez@csic.es)

S anders and Lehn groups reported the concept of Dynamic Combinatorial Chemistry (DCC) in the mid-1990s[1,2]. By using reversible chemical reactions, DCC establishes molecular networks under thermodynamic control that respond to external stimuli[3–5].

DCC systems that employ a protein to direct assemblies of small molecules at dynamic equilibrium are highly interesting. Huc and Lehn reported the use of carbonic anhydrase as a template proving its inhibition by a dynamic combinatorial library (DCL) of imines created in situ[6]. Since then, successful applications discovering novel enzyme inhibitors have been reported[7]. On protein-directed DCC experiments, one designs the system rather than the molecule allowing the protein to find its best ligand in situ[8–10].

The Neuronal Calcium Sensor 1 (NCS-1) is a high-affinity $Ca^{2+}$-binding protein predominantly expressed in neurons[11,12]. NCS-1 is a highly conserved protein[13] that regulates synaptogenesis, synaptic transmission and is critical for learning and memory[11,14–16]. The *Drosophila* Neuronal Calcium Sensor 1 (*d*NCS-1 or Frequenin-2) displays a large, concave hydrophobic crevice onto which the guanine exchange factor Ric8a binds[15,17]. The interaction between NCS-1 and Ric8a regulates synapse number and probability of neurotransmitter release, thus constituting a pharmacological target for synaptopathies[15,18]. NCS-1 contains a C-terminal dynamic helix (called H10) that works as a built-in competitive inhibitor and inserts into the crevice to prevent Ric8a binding (Fig. 1). In fact, inhibitors of this protein-protein interaction (PPI) target the NCS-1 crevice and stabilize the orientation that presents the helix H10 inside the crevice. This topology in turn, decreases synapse number and enhances associative learning in a Fragile X syndrome animal model[18,19]. Following the same reasoning, it would be tempting to hypothesize that an stabilizer of this PPI would permit to enhance synapse number and therefore constitute a pharmacological target of neurodegenerative diseases, where synapse number is abnormally low (Fig. 1)[20,21]. The low number of ligands reported for *d*NCS-1[18,19] makes the DCC approach attractive as a genuine discovery

tool for modulators able to unveil the mechanism to control neurotransmission[22] and synaptogenesis.

NMR spectroscopy has been reported as a particularly useful technique to analyze protein–ligand interactions (e.g., STD-NMR, tr-NOESY) and to understand reaction mechanisms using Ultrafast NMR (UF-NMR)[23,24].

Herein, we apply the DCC approach targeting *d*NCS-1 at low temperatures and physiological pH with an efficient catalyst to accelerate the DCL equilibration. UF-NMR technique is used to monitor in real-time the details of the acylhydrazone exchange process. Next, ligand-based NMR methods (STD-NMR, tr-NOESY and DOSY) in the presence of *d*NCS-1 are performed to get further insights into the interaction aspects of the chemical process. Moreover, the affinity of the amplified molecules is measured using fluorescence techniques and the modulation of the NCS-1/Ric8a interaction is tested in a protein-protein binding assay. These methodologies together with blood–brain barrier penetration assays, cell toxicity studies and ADME predictions permit to identify compound **3b** as the most effective molecule able to stabilize the NCS-1/Ric8a complex. Furthermore, the structure of the homologous *h*NCS-1 bound to **3b** is solved by X-ray diffraction to understand at the atomic level the basis of its ability to modulate the protein-protein interaction. Importantly, the therapeutic potential of compound **3b** is also assessed in vivo, showing that **3b** mediates the recovery of normal synapse number and improves the locomotor activity in a *Drosophila* model for Alzheimer´s disease.

## Results

**Acylhydrazone exchange catalyst at low temperature.** Most of the protein-directed DCC approaches reported to date have been tested under room temperature conditions[7,8]. However, in our case, besides the standard requirements such as compatibility with the biological target and short equilibration time, the reaction must occur at neutral pH and low temperatures to increase the stability time of *d*NCS-1.

To conduct acylhydrazone exchange at neutral pH, Greaney and coworkers used high concentrations of aniline as a nucleophilic catalyst in a protein-directed DCL[25]. The aniline catalyzed the acylhydrazone exchange through a Schiff-base intermediate. We started our DCL by reacting aldehyde **1** (Fig. 2a), with an excess of five acylhydrazides (**2a**–**2e**) at 4 °C in the absence and presence of the protein. Unfortunately, the required high concentrations of aniline interfered with the techniques employed for the analysis of protein–ligand interactions. Therefore, we studied different *p*-substituted aniline bases such as *p*-aminophenol, *p*-anisidine and *p*-phenylenediamine to compare their efficiency as nucleophilic catalysts, given the capacity of electron donating groups on *p*-position of the ring for increasing the basic character of the corresponding Schiff-bases (Fig. 2b)[26].

HPLC-MS was used to screen the proposed catalysts for the formation of the acylhydrazone **3b**. The reaction was performed at 4 °C in the presence and in the absence of the *p*-substituted aniline derivatives. The reactions were initiated by the addition of the aldehyde **1** and the formation of **3b** (Fig. 2c) was monitored over time. The resulting data were fit to a pseudo-second-order rate equation (see Supplementary Figs. 1, 2 and Supplementary Methods) and the kinetic parameters are summarized in Fig. 2d.

Under our experimental conditions, *p*-phenylendiamine and *p*-anisidine showed superior catalytic activity compared to aniline. Fig. 2c illustrates the time course of the reaction. In the absence of the catalyst (Fig. 2d), the half-time ($t_{1/2}$) of the reaction is 303 min ($K_{obs} = 0.61 \pm 0.02 \, M^{-1} \, s^{-1}$). As expected, aniline enhanced the rate of acylhydrazone formation reducing the $t_{1/2}$ from 303

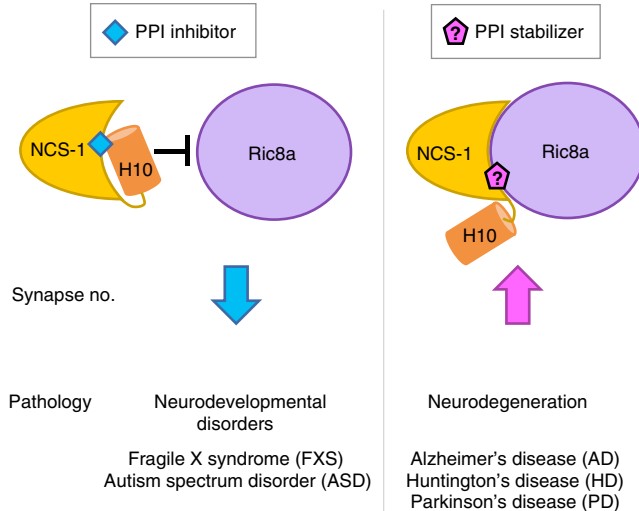

**Fig. 1** The complex between NCS-1 and Ric8a as a target for synaptopathies. Schematic representation of the regulation mechanism of the PPI target with small molecules. Examples of pathologies associated with an abnormal synapse number and the modulatory effect (decrease or increase in synapse number) exerted or expected by the small molecule modulators are also given. The key NCS-1 C-terminal helix is represented as an orange cylinder

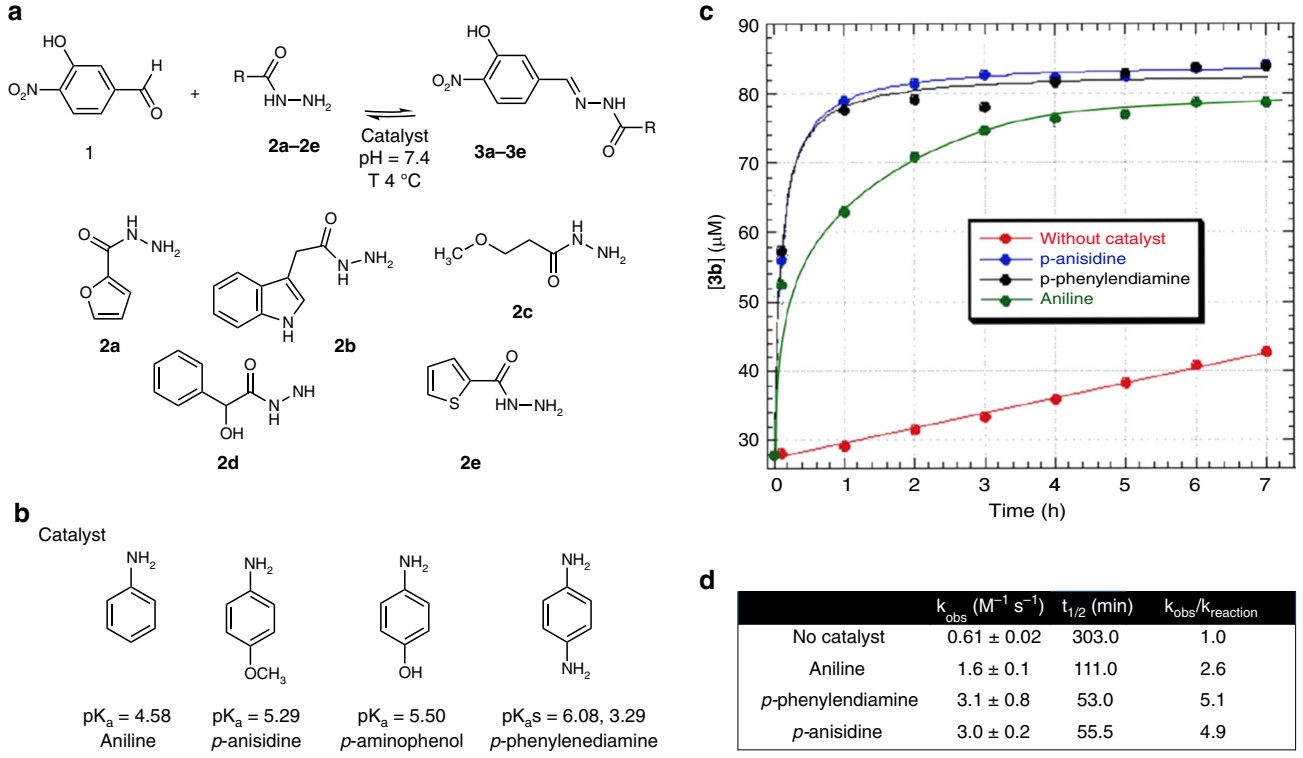

**Fig. 2** Aniline derivatives tested as DCL hydrazone exchange catalysts[27]. **a** DCL building blocks and library conditions at physiological pH and low temperature, **b** Aniline derivatives used as catalysts. **c** Time course formation of compound **3b** using an aldehyde concentration of 0.09 µM in 20 mM Tris buffer (pH 7.4), acylhydrazide **2b** (0.27 µM), $T = 4$ °C, 5% DMSO in the absence of the catalyst (red dots), and in the presence of 15 mM of aniline (green dots), p-anisidine (blue dots) and p-phenylendiamine (black dots). **d** Kinetic parameters ($K_{obs}$ and $t_{1/2}$) of acylhydrazone **3b** formation calculated for a pseudo-second-order rate equation in the absence or in the presence of different catalysts (Supplementary Fig. 1 and Supplementary Methods). Mean ± SD from three independent experiments. The right column shows the rate enhancement of catalysts relative to the uncatalyzed samples. p-aminophenol was discarded as it got quickly transformed into the quinone derivative. Source data are provided as a Source Data file

to 101 min ($K_{obs} = 1.6 \pm 0.1\,M^{-1}\,s^{-1}$). However, the $t_{1/2}$ for p-phenylendiamine and p-anisidine was highly reduced from 303 min to 53 min (p-phenylendiamine, $K_{obs} = 3.1 \pm 0.8\,M^{-1}\,s^{-1}$) and 55.5 min (p-anisidine, $K_{obs} = 3.0 \pm 0.2\,M^{-1}\,s^{-1}$), respectively. The reaction is completed after 2.5 h. Due to solubility reasons we decided to use p-anisidine instead of p-phenylendiamine as catalyst.

**Real-time acylhydrazone exchange mechanism.** NMR experiments were then conducted to monitor the dynamic acylhydrazone exchange in real-time and to confirm the proposed mechanisms in the absence (path i) and in the presence (path ii) of the catalyst at physiological pH (Fig. 3).

Initially, the products and intermediates participating in the reaction between aldehyde **1** and acylhydrazide **2b** were identified in the absence of the catalyst (Fig. 3a) using standard 1D-$^1$H-NMR spectra acquired in a sequential manner (Fig. 3b). Intermediate I (green) was identified by analyzing their $^1$H-NMR signals (the signal at $\delta$5.25 ppm corresponds to the H on the carbinolamine carbon while that at $\delta$7.79 ppm, represents the aromatic H *ortho* to the nitro group). These signals disappear as the final product is being formed. In fact, the formation of **3b** can be followed by the increasing presence of the imine-type $^1$H-NMR signal at $\delta$ 8.13 ppm (purple). Although the mentioned NMR signals of intermediate I and **3b** are already present in the initial recorded NMR spectrum, the aldehyde signals completely disappeared after 24 h. As expected, at physiological pH, the acylhydrazone formation is rate limited by the dehydration step.

Similar sequential 1D-$^1$H-NMR spectra were recorded to study the catalytic pathway adding **2b** to the mixture of the catalyst

(p-anisidine) and **1**. However, in this case, the intermediates could not be identified due to the higher speed of this process and to the severe overlapping of the NMR signals arising from the mixture. Therefore, we considered the use of 2D-NMR to get better signal dispersion.

The so-called ultrafast 2D-NMR (UF-NMR) method was employed, since it has been demonstrated that it may be used to monitor chemical reactions in-situ[23,24]. In particular, 2D-UF-TOCSY experiments were recorded to identify the intermediates formed upon adding **2b**, using a fast mixing device, into the NMR tube containing a solution of **1** and p-anisidine. The p-anisidine concentration was 0.5 equivalents with respect to **1** to be able to detect the three different p-anisidine states (free state, Schiff-base, Intermediate II).

Figure 3c shows a selection of different UF-TOCSY experiments recorded at different times (see video in the Supplementary Movie for the sequence of the five hundred UF-TOCSY recorded spectra and Supplementary Fig. 12). Initially, the NMR signals for the imine-type proton, the aromatic protons of the aldehyde and the p-anisidine, both taking part in the Schiff-base (see Supplementary Fig. 11) were readily identified (red). At 7 min, the signal of the imine proton of **3b** was already observable (purple), while the NMR signals of the protons at p-position from the released p-anisidine catalyst (blue) were evident. The process finished after 40 min, but it was not possible to confirm the presence of the Intermediate II. Furthermore, a small amount of Intermediate I (11%) was observed in the UF experiments as result of the coupling between aldehyde **1** and acylhydrazide **2b** (Supplementary Figs. 13, 14 and Supplementary Table 4).

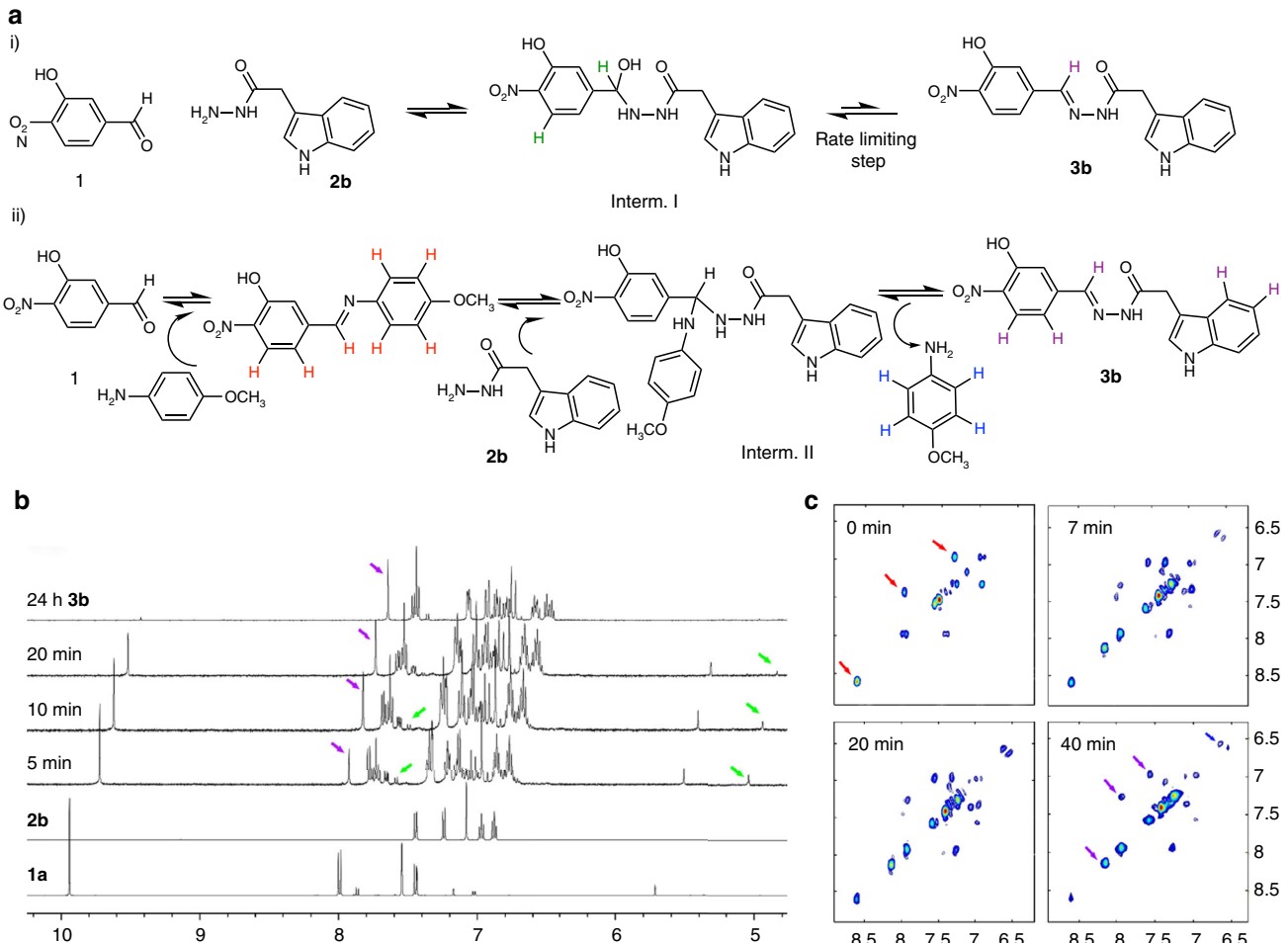

**Fig. 3** Acylhydrazone exchange reaction mechanism. **a** Formation of acylhydrazone **3b** in the absence (i) and presence (ii) of *p*-anisidine. **b** Real-time 1D $^1$H NMR series recorded as a function of time (only a small subset of the resulting spectra hereby shown) of the reaction of **1** (50 mM) and **2b** (50 mM) in Tris buffer D$_2$O/DMSO-$d_6$ (1:4) at 298 K (500 MHz). Colored arrows show the positions of specific signals from products and intermediates. The horizontal offset in ppm is 0.1. Note that there is a small fraction of the hydrated aldehyde in the spectra. **c** Plots of four selected UF-2D-TOCSY NMR spectra taken from the 500 experiments acquired (**1**:**2b**:*p*-anisidine at 1:1:0.5 in Tris buffer D$_2$O/DMSO-$d_6$ (1:4) at 298 K (500 MHz). Cross-peaks from the Schiff-base intermediate and the final step of formation of **3b** are depicted in red and purple respectively

Experiments with other acylhydrazides were also performed confirming these results.

**$d$NCS-1 dynamic combinatorial library**. After establishing *p*-anisidine as catalyst, the DCL approach was attempted by mixing aldehyde **1** (Fig. 4a) with five acylhydrazides (**2a**–**2e**) in the presence of $d$NCS-1. The DCL control was also performed in the absence of the protein. The selection of the aldehyde and the 5 acylhydrazides was based on previous DCL experiments in which the building blocks reactivity and their concentrations were carefully assessed to ensure the full solubility of the different components. The stability of the protein under the experimental conditions (DMSO tolerance and stability over time) was also tested using fluorescence and NMR techniques (Supplementary Figs. 15, 16 and Supplementary Methods). The equilibration was completed after 5 h (Fig. 4a) and the acylhydrazones were identified by HPLC-MS (Supplementary Figs. 4–8).

Aldehyde **1** could not be detected, indicating that it was continuously being sequestered as an acylhydrazone component. The reversibility of the DCL was evident, since an identical equilibrium distribution to that shown in Fig. 4a was obtained when two different starting points were employed.

The observed degree of amplification was **3b** > **3e** ≥ **3d** > **3a**. The precise composition of the DCL (with and without $d$NCS-1), was assessed by measuring the relative peak area (RPA). Indeed, the normalized change of RPA was used to quantify the protein influence in the final outcome (Supplementary Figs. 9, 10 and Supplementary Tables 1, 2 and 3)[28]. The presence of acylhydrazone **3c** was clearly reduced in presence of $d$NCS-1 indicating a lack of significant affinity for $d$NCS-1. Note that compounds **3a**–**3e** can exist as E/Z isomers of the C = N bond. Quantum mechanics calculations of the geometries for the E/Z stereoisomers of **3a**–**3e** revealed that isomer E is preferred; both in vacuum and in water (Supplementary Table 6). Interestingly, the calculated pK$_a$ for the acylhydrazone NH (8.0 and 8.5 for Z and E isomers, see Supplementary Figs. 19 and 20) of compound **3b** shows the acidic nature of this NH proton, which strongly suggests that the isomerization from the Z to the most stable E isomer may easily occur in the reaction medium at pH 8.

DOSY-NMR and tr-NOESY-NMR experiments were also recorded to follow the exchange process. The obtained DOSY spectra in the presence of $d$NCS-1 revealed that the formed products displayed larger diffusion coefficients than the initial components, in agreement with their larger size increase (Fig. 4b).

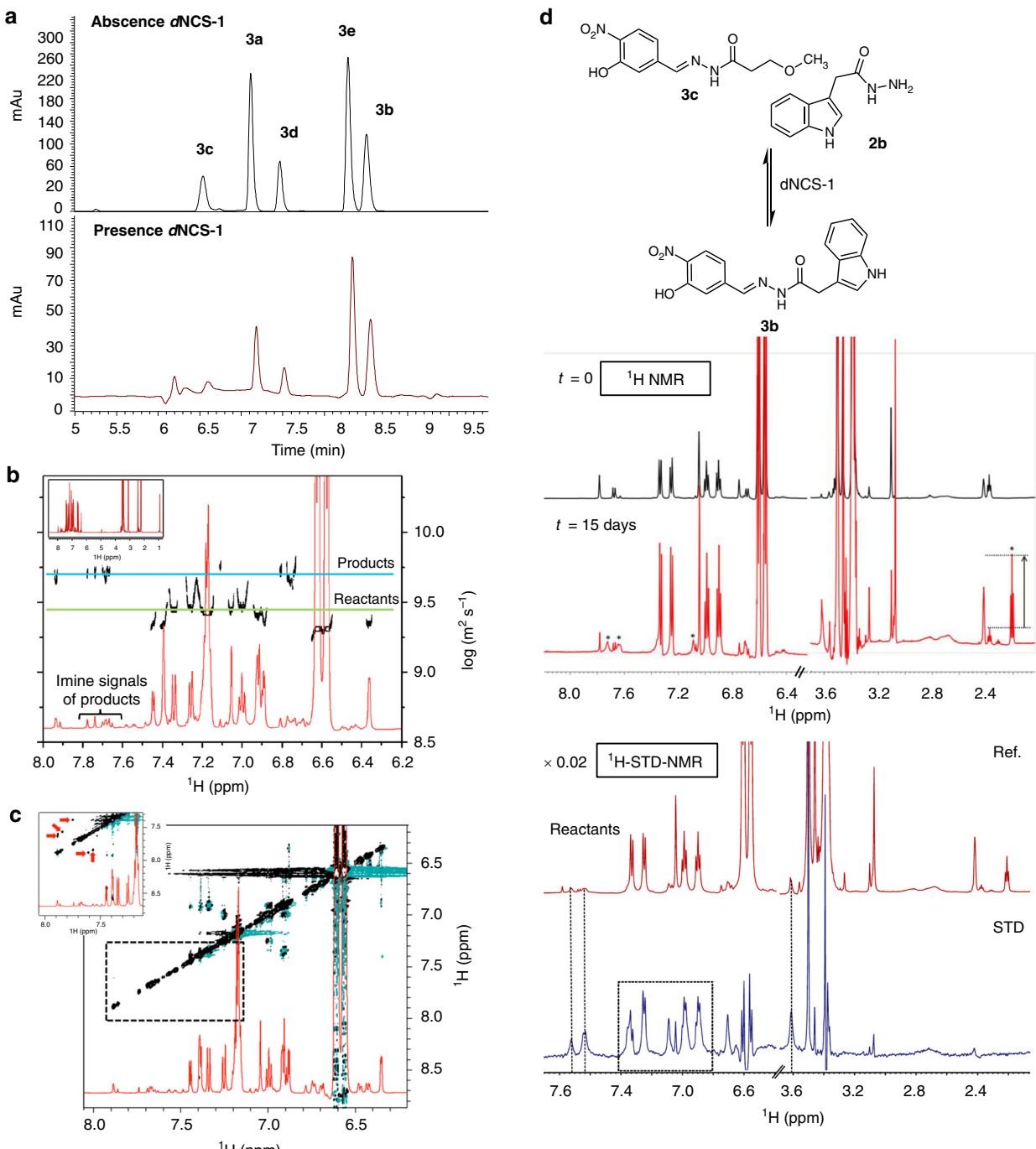

**Fig. 4** HPLC-MS and NMR studies of the full dynamic combinatorial library. **a** DCL chromatograms after 5 h in the absence and in the presence of *d*NCS-1. Conditions: aldehyde **1** (1.2 μL, 50 mM), **2a**-**2e** (3.6 μL, 50 mM), catalyst (1 μL, 12 M), *d*NCS-1:**1** [1:1], Tris buffer (20 mM, pH 7.4), 1 mM CaCl₂, 0.5 M NaCl, 1 mM DTT, T = 4 °C, 2% DMSO. DCC experiments were carried out in triplicate. **b** 1D-¹H NMR spectrum (red) of the mixture and DOSY experiment (black) of the DCL in the presence of *d*NCS-1 (281 K, 600 MHz). **c** Tr-NOESY spectrum of the DCL mixture in the presence of *d*NCS-1 (mixing time 200 ms, 281 K, 600 MHz). Amplification of the region δ 7.2–8.0 ppm. The negative transferred NOE cross-peaks corresponding to intramolecular NOEs of the products while bound to the protein are highlighted with red solid arrows. **d** ¹H-NMR spectrum of the sample with acylhydrazide **2b** and acylhydrazone **3c** in the presence of *d*NCS-1 at different reaction times and up to 15 days (281 K, 600 MHz). The new signals that reveal the formation of **3b** (aromatic region) and **2c** (aliphatic region) are marked with stars. ¹H-STD-NMR (blue) and off-resonance (red) NMR spectra of the mixture **2b** + **3c** + *d*NCS-1 (281 K, 600 MHz) after 15 days

Moreover, the presence of protein-bound products was further assessed by the presence of negative cross-peaks for the acylhydrazones in the tr-NOESY experiments, while the reactants and the catalyst only displayed positive and zero-quantum cross-peaks (Fig. 4c).

Additional probe of the existence of a protein template effect was extracted from the NMR analysis of the evolution of the mixture of **3c** (extremely weak or non-binder) and **2b** with *d*NCS-1 (Fig. 4d). The ¹H-NMR spectra revealed the presence of signals at the aromatic region assigned to **3b** as well as a triplet at

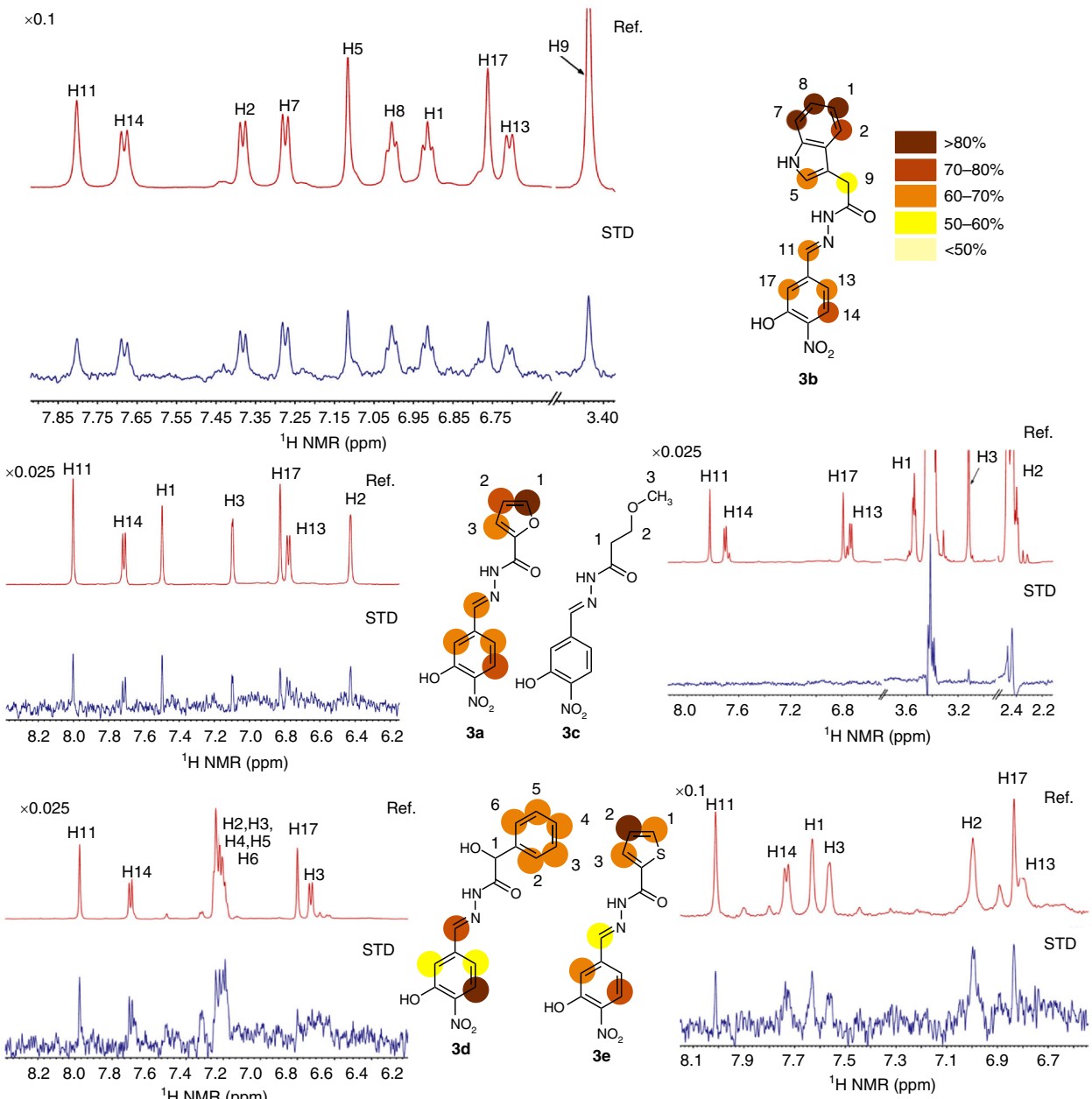

**Fig. 5** $^1$H-STD-NMR spectra of **3a**–**3e** in the presence of *d*NCS-1. STD (blue) and off-resonance (red) NMR spectra acquired for compounds **3a**–**3e** (1 mM) in the presence of 10 μM *d*NCS-1 (600 MHz, 281 K). Relative STD intensities are coded according to the color scale shown. The 100% STD signal corresponds to the resonance showing the highest intensity in each case and all other STD signals are calculated accordingly

δ 2.2 ppm. corresponding to **2c**, the acylhydrazide precursor of **3c**. Thus, *d*NCS-1 induces the synthesis of **3b**, a *d*NCS-1 ligand, at the expense of **3c**, which is not bound to the protein.

**NMR binding studies and compounds epitope mapping**. The analysis of the STD-NMR spectra[29] further identified four acyl-hydrazones as *d*NCS-1 binders (Fig. 5), while **3c** was not recognized. Compound **3b** displayed the largest STD intensities. Moreover, the STD analysis permitted to map its binding epitope, revealing structural details of the *d*NCS-1/**3b** binding mode.

**NCS-1 affinity and NCS-1/Ric8a complex modulation**. Fluorescence-based experiments with **3a**–**3e** were carried out to estimate their affinity to *d*NCS-1. As shown in Fig. 6a, the binding of **3a**, **3b** and **3d** quenches the fluorescence of tryptophans W30

and W103 located in the *d*NCS-1 hydrophobic cavity[17]. A similar effect was observed when Chlorpromazine (CPZ), an anti-psychotic drug and a well-known *d*NCS-1 binder[18], was used as control. The apparent $K_d$ for **3a** ($K_d = 32 \pm 2$ μM), **3b** ($K_d = 43 \pm 6$ μM) and **3d** ($K_d = 61 \pm 9$ μM) were slightly larger than that of CPZ ($K_d = 12 \pm 2$ μM), suggesting the existence of a similar binding affinity of all these molecules to *d*NCS-1. Compound **3c** did not show affinity for *d*NCS-1 while the limited solubility of **3e** under the experimental conditions precluded the acquisition of the data. Therefore, **3c** and **3e** were discarded for further studies.

Binding assays with NCS-1 and Ric8a in co-transfected HEK cells were carried out to study the modulation effect of compounds **3a**, **3b** and **3d** in the protein-protein interaction (Fig. 6b). CPZ, a reported mild inhibitor of the NCS-1/Ric8a interaction[18], was also included for comparison purposes. Interestingly, our data showed that compounds **3b** and **3d**

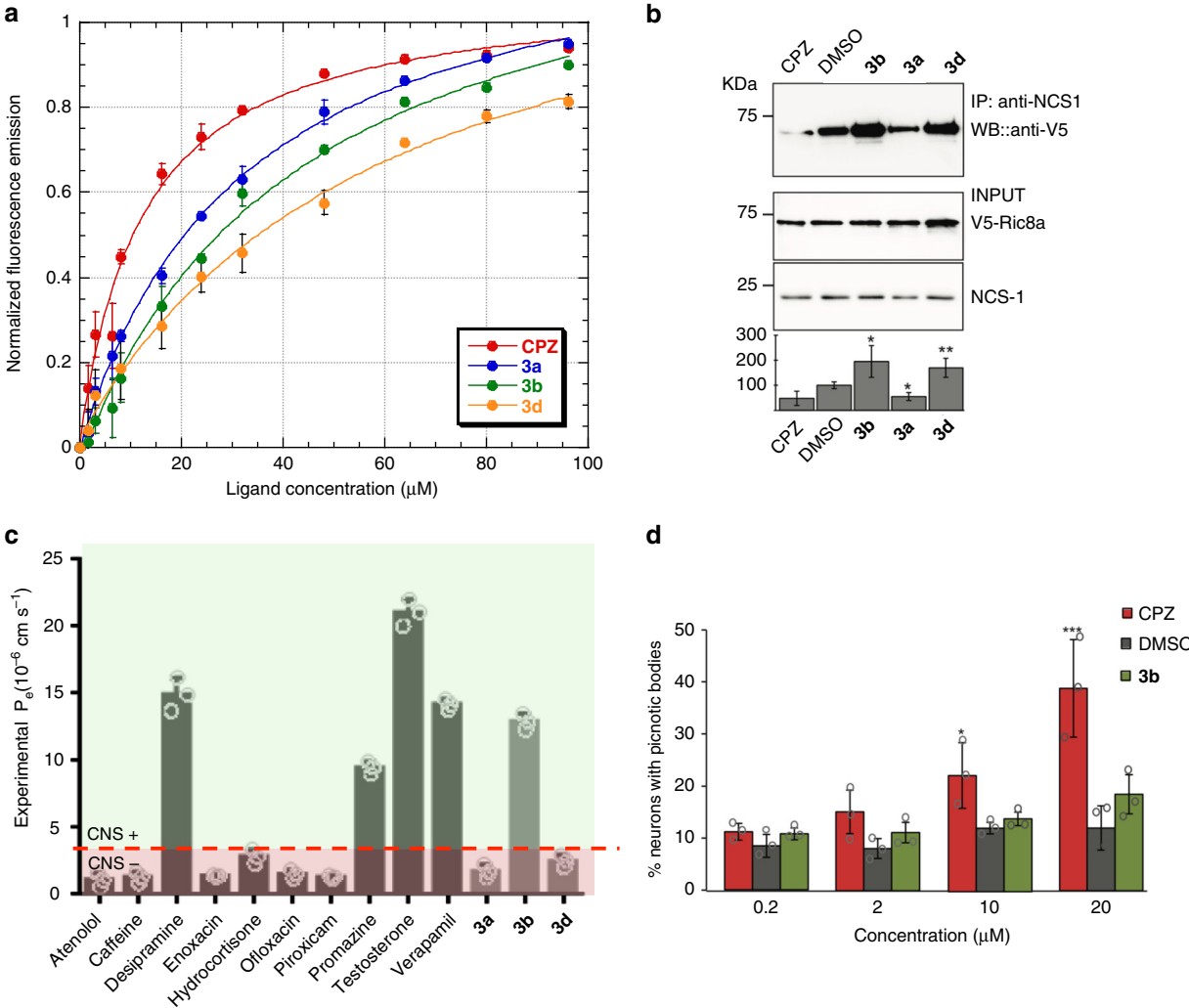

**Fig. 6** Protein–ligand binding and toxicity studies. **a** Representation of the fluorescence emission of $Ca^{2+}$ loaded $d$NCS1 at increasing concentration of ligand (**3a**–**3d** or CPZ). Mean ± SD from three independent experiments. The curves represent the least squares fitting of the experimental data to a 1:1 stoichiometry. To properly compare the different curves, intensities were normalized and represented. CPZ-$d$NCS-1 (red dots); **3a**-dNCS1 (blue dots); **3b**-dNCS-1 (green dots); **3d**-dNCS-1 (orange dots). **b** Co-IP binding assay of human NCS-1 and V5-tagged Ric8a in transfected HEK cells in the presence of CPZ, **3b**, **3a**, **3d** (20 μM) and the vehicle DMSO. Input represents 1/10 cell lysates before IP. Quantifications of each lane from four experiments are shown below the blots. Bars represent percentage of NCS-1/ Ric8a binding (mean ± SD) normalized to DMSO. Note the reduced binding in the presence of CPZ or **3a** and the strong binding with **3b** or **3d**, comparisons are with DMSO which represents basal binding levels (100%). **c** PAMPA in vitro permeability ($P_e$) plot of compounds **3a**, **3b** and **3d** and the reference drugs. CNS + (green) for $P_e > 4.47 \times 10^{-6}$ cm s$^{-1}$, CNS- (red) for compounds with $P_e < 4.47 \times 10^{-6}$ cm s$^{-1}$. Mean ± SD from three independent experiments. **d** Cell toxicity assay of CPZ, **3b** and the vehicle DMSO as control. Mean ± SD from three independent experiments. Cortical neurons from E14 wild-type mice were treated for 24 h with 0.2, 2, 10, 20 μM of CPZ (red), compound **3b** (green) or the same volume of the vehicle DMSO. Then, the percentage of picnotic bodies over the total nuclei was analyzed. Mean ± SD from three independent experiments. Paired two-tailed Student's test ***P<0.001; **P < 0.01; *P < 0.05. Source data are provided as a Source Data file. IP immunoprecipitation, WB western blot

promote the stabilization of the NCS-1/Ric8a interaction whereas **3a** is an inhibitor similar to CPZ.

**In vitro permeability**. Taking into account that one of the main difficulties in treating central nervous system diseases is the drug's capacity to cross the blood–brain barrier (BBB), the ability of compounds **3** (**a**, **b**, and **d**) to enter into the brain by passive diffusion was evaluated in a Parallel Artificial Membrane Permeation Assay (PAMPA methodology, Supplementary Fig. 18 and Supplementary Methods)[30]. The PAMPA methodology is a high-throughput technique to predict passive permeability through biological membranes that employs a brain lipid porcine as membrane. The in vitro permeabilities ($P_e$) of **3a**, **3b** and **3d** and ten commercial drugs were then determined. Compounds with $P_e > 4.47 \times 10^{-6}$ cm s$^{-1}$ are able to cross the BBB by passive

diffusion. As a result, compound **3b** can be classified as CNS + with a permeability of $12.9 \pm 0.8 \times 10^{-6}$ cm s$^{-1}$. In contrast, **3a** and **3d** did not show good permeability values (Fig. 6c, Supplementary Table 5).

**In silico physicochemical parameters and neuron viability**. In addition, in silico evaluation of Absorption, Distribution, Metabolism and Excretion (ADME) descriptors such as log $P_{o/w}$ (pH-independent partition coefficient) and log D (pH-dependent partition coefficient) were predicted for **3b**, obtaining 2.58 and 2.04, respectively at pH = 8 (Supplementary Table 7). This is in agreement with optimal log $P_{o/w}$ values (as an indicator of brain-blood partitioning) of 1.5–2.5 for drugs targeting CNS[31]. Furthermore, the aqueous solubility (log S) of **3b** was also calculated

**Table 1 Diffraction data collection and refinement statistics**

| Data collection | |
|---|---|
| Space group | P2$_1$ |
| Cell dimensions | |
| $a$, $b$, $c$ (Å) | 53.73, 55.60, 77.72 |
| $\alpha$, $\beta$, $\gamma$ (°) | 90.00, 94.97, 90.00 |
| Resolution (Å) | 42.35–1.78 (1.82–1.78)[a] |
| $R_{pim}$ | 0.045 (1.128) |
| CC$_{1/2}$ | 0.998 (0.355) |
| $I$ / σ$I$ | 8.6 (0.7) |
| Completeness (%) | 99.6 (199.8) |
| Wilson B-factor | 30.33 |
| Multiplicity | 3.4 (3.3) |
| Refinement | |
| Resolution (Å) | 42.35–1.78 (1.80–1.78) |
| No. reflections | 43787 |
| $R_{work}$ / $R_{free}$ | 21.35/23.18 (39.26/42.21) |
| Asymmetric unit content | |
| No. atoms | 6459 |
| Protein (residue range) | 2 (3–189 and 3–188) |
| **3b**/PEG/DMSO/Acetate | 1/7/1/1 |
| Calcium/Sodium ions | 6/2 |
| Water molecules | 179 |
| B-factors (Å$^2$) | |
| Protein | 49.39 |
| Ligand/ion | 63.55 |
| R.m.s. deviations | |
| Bond lengths (Å) | 0.014 |
| Bond angles (°) | 1.303 |

[a]Diffraction data collected from one crystal (Values in parentheses are for highest-resolution shell)

yielding values of −4.183/−4.675, similar to those obtained for other CNS drugs (see Supplementary Table 8).

Finally, cell toxicity in neurons was quantified as percentage of picnotic bodies for **3b** (Fig. 6d). There were no significant differences in **3b** treated cells with those obtained with the same amount of the drug vehicle, DMSO. Our results suggest a physiological effect of compound **3b** without affecting neuron viability.

In light of these results, compound **3b** was chosen as candidate to understand the binding properties and to study the in vivo effect as a promising hit compound.

**The crystal structure of *h*NCS-1 bound to 3b**. To understand the activity of **3b** as an stabilizer of the NCS-1/Ric8a interaction, the crystal structure of the Ca$^{2+}$ bound *h*NCS-1/**3b** was solved at 1.78 Å resolution (PDB code 6QI4, Table 1). Crystals belonged to the monoclinic P2$_1$ space group. The asymmetric unit (AU) contained two *h*NCS-1 molecules with an RMSD for all atoms of 1.28 Å. The feature-enhanced and the $2F_o − F_c$ electron density maps, together with different map calculations (see Fig. 7a and Supplementary Fig. 17) allowed the unambiguous modelling of **3b** bound to the hydrophobic crevice of one of the two independent *h*NCS-1 molecules of the AU, while the second *h*NCS-1 molecule only showed a PEG molecule at the **3b** equivalent position (Fig. 7). Interestingly, **3b** targets the same region as other inhibitors (Fig. 8)[18,19], displaying a contact area of 303.4 Å$^2$[32].

The amino acids participating in **3b** recognition are: W30 and D37 (helix H2), defining the upper wall of the cavity (Fig. 7c). The base of the cavity is formed by F72 and V68 (helix H4), F48 (helix H3) and W103 (helix H6). As lateral walls: F85, L89 and T92 (helix H5) and opposite to it it, I51, F55 and Y52 (helix H3). The indole group of **3b** is stabilized with π-π interactions with W30 and F85, weak water-mediated H-bonds with D37, and hydrophobic interactions with L89 and I51. The acylhydrazone

oxygen of **3b** is forming a strong H-bond with a water molecule in the upper part of the cavity. Nitrogen atom N2 is stabilized with van der Waals contacts with F48 and nitrogen N3 with Y52 and F55, being the latter mediated by a water molecule (w158). Furthermore, hydrophobic interactions are observed between C11 and F48 and F72. Interestingly, the electron density map showed that the NCS-1 bound **3b** molecule only displays the $E$ geometry, the QM-predicted and most stable isomer (Supplementary Table 6). Nevertheless, since **3b** is present in solution as a mixture of $Z/E$ isomers, the molecular recognition event takes place with a conformational selection process. In addition, the 2-hydroxy-3-nitrophenyl ring perpendicular to the surface interacted with V68 and Y52 and F72, W103 and T92. The **3b** most implicated atoms in these interactions are C13, C14 and C17. It is important to note that the interactions observed in the *h*NCS-1/**3b** crystal structure match the STD-NMR epitope mapping (Fig. 5).

When comparing the **3b**-free and bound *h*NCS-1 structures found in the asymmetric unit, a rearrangement takes place to allow ligand recognition: helix H3 shifts and D37 carboxyl reorients to establish weak H-bonds between a group of water molecules and **3b** indole group (Fig. 7d). Indeed, I52 side chain changes to permit the positioning and interaction of water w155. Furthermore, T92 side chain, that shows double conformations in the absence of **3b**, fixes its conformation in the presence of **3b** through a H-bond with a water molecule, which permit to establish hydrophobic contacts with the 2-hydroxy-3-nitrophenyl ring.

The comparison of *h*NCS-1/**3b** structure with the reported structures of *d*NCS-1 bound to strong (FD44, IGS-1.76)[18,19] and mild (CPZ) inhibitors (Fig. 8) shows that **3b** indole group is placed upper in the cavity enabling D37 participation. Moreover, **3b** does not form strong H-bonds with T92 and Y52 or contact the helix H10, as inhibitors do (Fig. 1 and Fig. 8b). Particularly, the binding of PEG molecules to the C-terminal part of the crevice stabilizes the helix H10 outside and parallel to it (Figs. 7b, 8a). While the strong inhibitors (Fig. 8a–b, d) use apolar rings to properly contact L182 and L184 and stabilize the helix H10 inside the crevice, compound **3b** locates at L182 and L184 interaction region the nitro and hydroxyl groups, hindering the stabilization of the helix H10 inside. Therefore, our data suggest that **3b** stabilizes the NCS-1/Ric8a complex keeping the helix H10 out of the crevice, and promoting the entrance and recognition of Ric8a.

Finally, the structural comparison of the PPI modulators indicate that these compounds can be divided in two parts: i) an aromatic region that targets the molecules to an aromatic-enriched area of the NCS-1 crevice and confers affinity (highlighted in Fig. 8d), and ii) a variable region that confers function: inhibition or stabilization. Inhibitors need long-enough hydrophobic moieties that reach the helix H10 for interaction, and stabilizers need polar groups that hinder the helix insertion (Fig. 8d).

**Compound 3b in a *Drosophila* model of Alzheimer's disease**. As we have established that compound **3b** stabilizes the NCS-1/Ric8a interaction and given the reported effects of this interaction on regulating synapse number and synapse function[15,16,18], we assayed **3b** on an in vivo model of synaptopathy, where synaptic loss is a primary hallmark of disease[20,21]. The expression of synaptotoxic amyloid aβ42arc in motor neurons leads to a reduction in the number of synapses with respect to normal age-matched neuromuscular junctions[33]. Moreover, the expression of amyloid peptides in *Drosophila* neurons displays various symptoms reminiscent of Alzheimer's disease including defective locomotion, memory loss or reduced longevity[34].

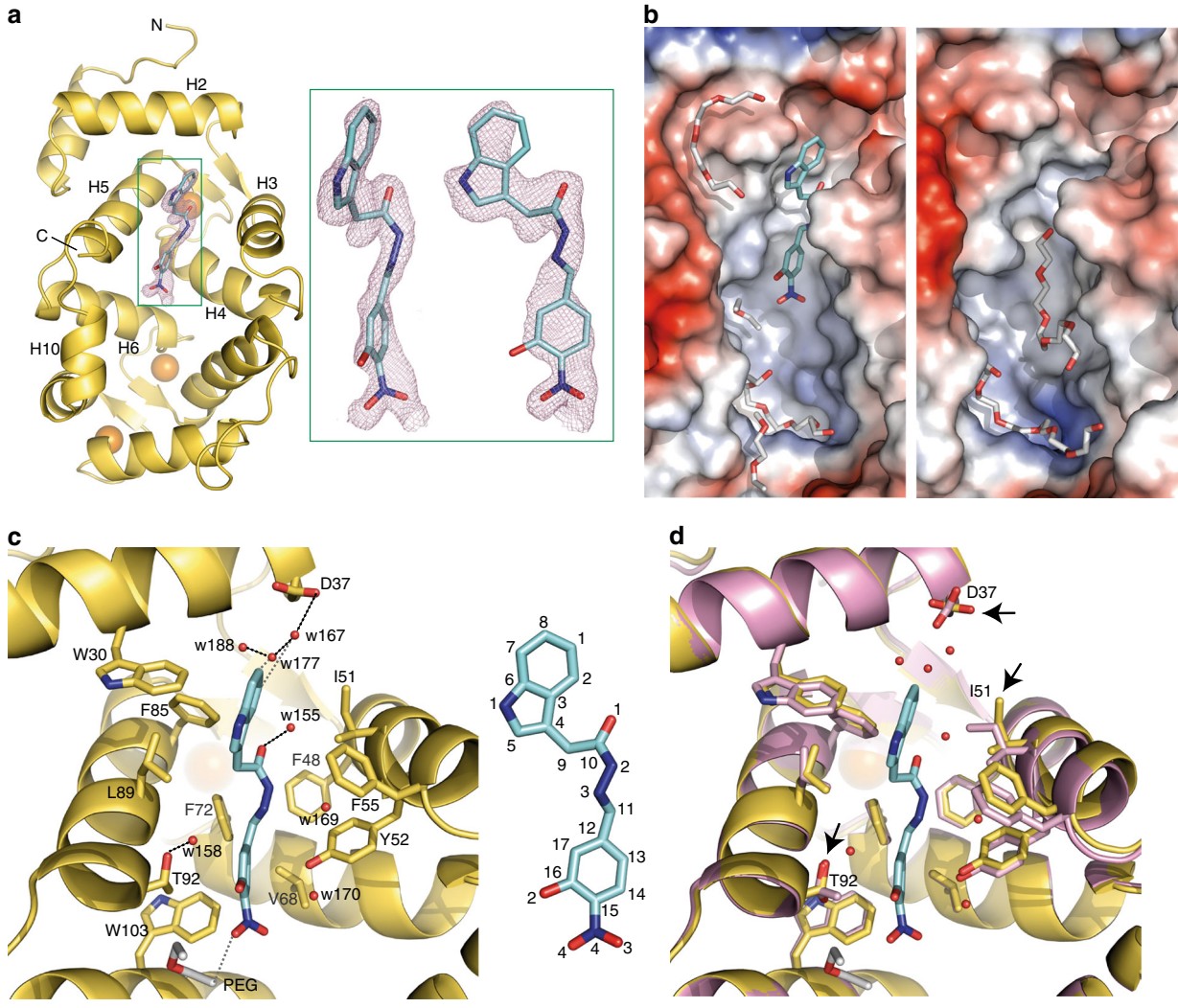

**Fig. 7** Structure of the $Ca^{2+}$ bound $h$NCS-1/**3b** complex. **a** Ribbon representation of $h$NCS-1 bound to $Ca^{2+}$ (orange spheres) and **3b** (cyan sticks). The calculated feature-enhanced map at the **3b** region is depicted in pink at $1.4\sigma$ level, and two zoomed-in views are shown (green square). **b** Electrostatic potential molecular surface representation of the two independent $h$NCS-1 molecules found in the AU showing the PEG content of the hydrophobic crevices (light grey sticks) and **3b**. **c** Detailed view of the residues (side chains as yellow sticks) and molecules recognizing **3b**. Strong and weak H-bonds are shown as black and grey dashed lines, respectively. **3b** atom numbering is represented. **d** The superimposition of the ligand-bound (yellow) and the ligand-free (pink) $h$NCS-1 molecules found in the AU. Arrows indicate the residues that suffer important rearrangements

Aβ42arc overexpressing flies and the corresponding control (LacZ expression) were fed with **3b** or the solvent, DMSO, throughout all life cycle (Fig. 9 and see Methods section). The data confirmed that synapse counting was reduced in aβ42arc[33], but this pathological phenotype was largely suppressed in the **3b** feed flies. By contrast, **3b** and its solvent DMSO showed no effect on the control flies (Fig. 9a).

To measure the physiological impact of this synaptic recovery, we evaluated fly locomotor activity. As described previously, overexpression of human aβ42arc peptide leads to severe locomotor dysfunction starting at day 15–20 post-eclosion[35]. Remarkably, the locomotor deficit was recovered by **3b** feeding (Fig. 9b). Furthermore, the statistical analysis of the data does not reveal a significant difference in the locomotor activity of control flies fed with **3b** vs. DMSO.

### Discussion

Adaptability is the essence for evolution and guides the emergence of diverse chemical structures. Modulators of protein-protein interactions are relatively rare. We designed a dynamic

reversible system from one aldehyde and five acylhydrazides able to uncover an unexpected protein-protein interaction stabilizer.

The reversible chemistry chosen, acylhydrazone exchange, was prepared to work at low temperatures and neutral pH using $p$-anisidine as a catalyst broadening its range of application to other biological targets. Moreover, ultrafast NMR experiments have allowed the detection of the carbinolamines (hemiaminals) intermediates and could successfully be applied to determine the mechanism of C=N double bonds formation of pyrazoles[37] and isoxazoles[38].

The calcium sensor protein NCS-1 has been proved to be an excellent DCL template directing the library to the synthesis of compound **3b**, the protein-protein interaction enhancer of the NCS-1/Ric8a complex ever reported, still with a moderate binding affinity. Nevertheless, detailed NMR and X-ray studies have shed light on the structural and chemical requirements to stabilize the NCS-1/Ric8a complex.

We had previously shown how the interaction of NCS-1 and Ric8a emerged as a potential therapeutic target for diseases affecting synapses, due to its role in regulating synapse number

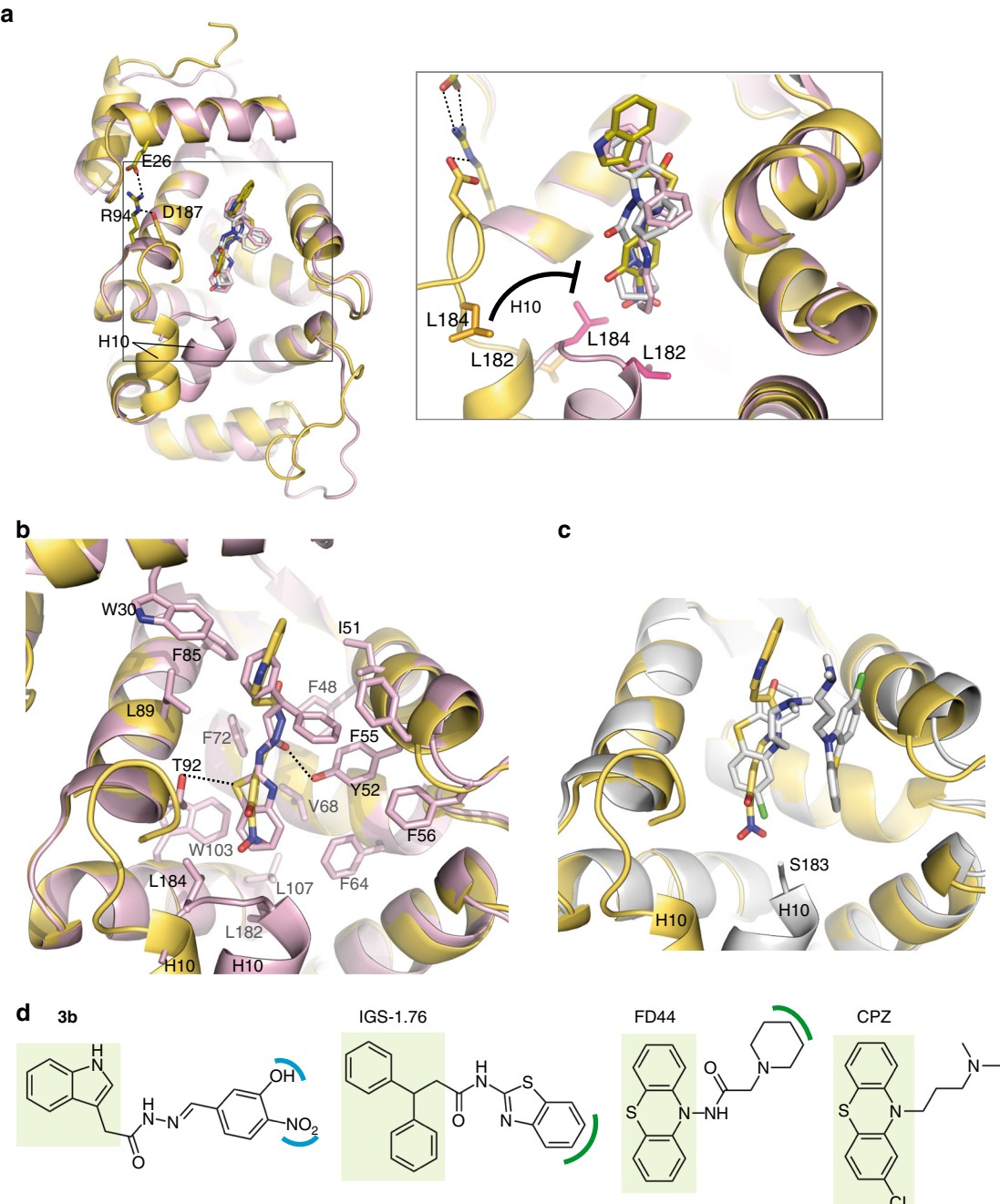

**Fig. 8** Structural analysis of the NCS-1/Ric8a small molecule regulators. **a**, **b** Structural comparison of **3b** with strong inhibitors. Superimposition of the structure of *h*NCS-1/**3b** complex (yellow ribbons/sticks) with that of *d*NCS-1/IGS-1.76 (pink ribbons/sticks) (PDB code 6epa)[19]. The structure of *d*NCS-1/ FD44 has also been superimposed[18], but only the small compound is represented (white sticks). The grey square represents a rotated zoomed-in view to visualize the ligands (**3b**, IGS-1.76 and FD44) and helix H10. **b** The residues contacting IGS-1.76 are shown. Strong H-bonds of IGS-1.76 with Y52 and T92 are displayed with black dashed lines. **c** Structural comparison of **3b** with a mild inhibitor. Superimposition of *h*NCS-1/**3b** complex (yellow) with *d*NCS-1/ CPZ (white) (PDB code 5g08) structures. Under the crystallization conditions, two different CPZ conformations were modeled. One of the conformations binds to the same site as compound **3b**. However, CPZ hydrophobic tail is not efficient enough in contacting helix H10 C-terminal end, which was found disordered in the crystal (from residue 184 to the end), and therefore the inhibition is mild[18]. **d** 2D structures of the regulatory molecules represented in (**a**, **c**). The aromatic region conserved in all PPI regulators is squared in light green. The region of FD44 and IGS-1.76 implicated in an efficient interaction with helix H10 are highlighted in green. The **3b** polar groups, sharing the same position, are highlighted in blue

and neurotransmitter release[11,15,16,18]. In this context, the in vivo results show that **3b**-mediated stabilization of the NCS-1/Ric8a complex, indeed increases the number of synapses to normal levels, exclusively in the presence of a synaptic pathology, which is an essential requirement for any treatment directed to synapses.

Therefore, compound **3b** constitutes a promising prototypic probe for further research in the treatment of neurodegenerative disorders such as Alzheimer's, Huntington's or Parkinson's diseases characterized by a decrease in the number and efficacy of synapses that precedes neuronal death.

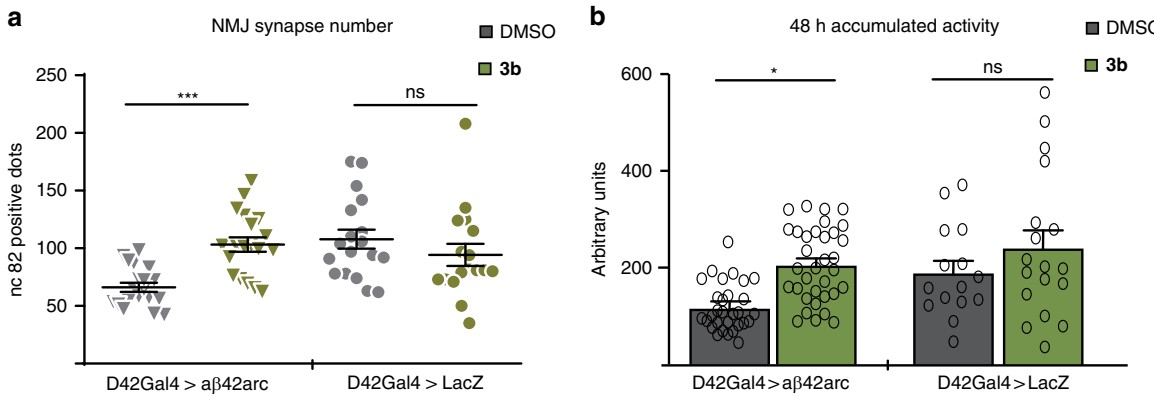

**Fig. 9** In vivo effects of compound **3b**. Flies with motoneuron overexpression of the human aβ42arc (arctic mutation) (D42Gal4 > aβ42arc) or mock expression (D42Gal4 > LacZ) were fed with **3b** (100 μM) or same volume of vehicle DMSO. **a** Twenty-day old adult abdominal motoneurons were analyzed and synapse number (nc82-positive spots) of the same motoneuron in different flies (16–19) were determined using Imaris, over the confocal 1 μm stacks. Data are plotted in graphs, where each grey triangle (aβ42arc flies fed with vehicle) and green triangle (aβ42arc flies fed with **3b**) or each grey circle (control flies fed with vehicle) and green circle (control flies fed with **3b**) represents one value. Horizontal line represents mean ± SEM. Data are analyzed statistically with unpaired two-tailored Student's $t$ test; ***$P < 0.001$. **b** Locomotor activity of individual 15 days old flies was recorded for 4 days in *Drosophila* Activity Monitors (DAM2, Trikinetics)[36], the total number of beam breaks per hour during two consecutive days was analyzed (the activity of the first two days is considered the habituation period and is discarded). Mean ± SEM of three independent experiments with 5–12 flies per condition each, were plotted and analyzed statistically with paired two-tailed Student's $t$-test, *$P < 0.05$. Source data are provided as a Source Data file, ns non significant

## Methods

**Catalysis of reversible acylhydrazone formation**. The kinetic experiment was performed adding aldehyde **1** (1.2 μL, 50 mM, DMSO), acylhydrazide **2b** (3.6 μL, 1.5 M, DMSO), the catalyst (p-anisidine, p-phenylendiamine, aniline) or control DMSO (1.0 μL) in buffer Tris (pH 7.4, 640 μL, 20 mM, 0.5 M NaCl, 1 mM CaCl₂, 1 mM DTT) at 4 °C at 15 mM and 50 mM catalyst concentration. 5% of DMSO was present in the final mixture. The absorbance data of **1** and **3b** were measured for seven hours by HPLC. The data were collected and treated by using a least squares algorithm to fit the equation for pseudo-second-order (Supplementary Figs. 1, 2 and Supplementary Methods)

**UF-NMR experiment**. A solution of aldehyde **1** (125 μL, 0.2 M, DMSO-$d_6$) and of p-anisidine (2 μL, 6 M, DMSO-$d_6$) were added to a mixture of DMSO-$d_6$ (273 μL) and Tris buffer D₂O (100 μL), in a 5 mm NMR tube. Inside of a NMR tube was assembled a fast mixing device for adding the acylhydrazide **2b** (50 μL, 0.5 M, DMSO-$d_6$) (Supplementary Methods).

**DCL preparation**. Aldehyde **1** (1.2 μL, 50 mM, DMSO), the five acylhydrazides **2a–2e** (5 × 3.6 μL, 50 mM, DMSO), p-anisidine (1 μL, 12 M, DMSO) and buffer 20 mM Tris, 0.5 M NaCl and 1 mM CaCl₂, 1 mM DTT at pH 7.4 (750 μL) in 3.3 % DMSO. The mixture was stabilized in 5 h at 4 °C. HPLC analysis was performed.

**Protein-directed-DCL**. Aldehyde **1** (1.2 μL, 50 mM, DMSO), the five acylhydrazides **2a–2e** (5 × 3.6 μL, 50 mM, DMSO), p-anisidine (1 μL, 12 mM, DMSO) and $d$NCS-1 in buffer 20 mM Tris, 0.5 M NaCl and 1 mM CaCl₂, 1 mM DTT at pH 7.4 (66.7 μM, 750 μL, 1 eq.). The experiment is in 3.3% DMSO. The mixture was stabilized for 5 h at 4 °C. Then, $d$NCS-1 was removed by ultrafiltration using an Amicon Ultra filter (0.5–10 KDa). HPLC analysis was performed and the traces were compared with the blank composition.

**Synthesis of acylhydrazones 3a–3e**. See Supplementary Fig. 3, Supplementary Methods and Supplementary Discussion.

**STD-NMR experiments**. The experiments were performed using a deuterated Tris (pH 7.9) buffered solution with an aliquot of DMSO-$d_6$. A sample containing 10 μM of $d$NCS-1 and 1 mM of DCL species was prepared. The STD experiments were acquired at 281 K on a Bruker Avance 600 MHz spectrometer equipped with a cryoprobe. The saturation frequency was set at δ −0.5 ppm (aliphatic region) and the saturation time was 2 s. A spin lock filter was applied to minimize signals from $d$NCS-1. The same conditions were used for the acquisition of the STD-NMR spectra of the individual products with $d$NCS-1.

**tr-NOESY and DOSY experiments**. The experiments were performed on the same samples and spectrometer, at 281 K. The tr-NOESY mixing time was fixed at 200 ms. The DOSY experiments were acquired with 16 gradient increments to a final intensity decay of 90%.

**Quantum mechanics calculations**. Geometry optimization and energy calculation of the stereoisomers **3a–3e** and conjugated base from **3b**. Supplementary Figs. 19 and 20, Supplementary Tables 6–8.

**Fluorescence experiments**. See Supplementary Methods.

**Co-Immunoprecipitation assays and western blotting**. $h$NCS-1 and V5-tagged $h$Ric8a were co-transfected into HEK293 cells (Dharmacon) using Lipofectamine 2000 (Invitrogen)[13]. After 24 h, after transfection, DMSO alone or the compounds dissolved in DMSO (20 μM) were added to the culture cells. Then, 24 h after transfection, cells were lysed with Lysis buffer (150 mM NaCl, 1.0% Nonidet P-40, 50 mM Tris, pH 8.0) in the presence of the compounds (20 μM), whose concentration was maintained throughout the immunoprecipitation assay. Precleared lysates were incubated overnight (12 h) at 4 °C with mouse anti-NCS-1 (1:500; Cell Signaling). Samples were subsequently incubated overnight with Protein-G-Sepharose (Sigma-Aldrich), washed and eluted from the Sepharose. Samples were analyzed by Western blot following standard procedures. The amount of V5-tagged $h$Ric8a bound to $h$NCS-1 was revealed by V5 antibody (1:1000, Invitrogen). 1/10 cell lysates before IP, were run in a Western blot and the NCS-1 input (anti NCS-1, Cell Signalling 1:2000) and Ric8a input (anti V5, Invitrogen, 1:2000) were then revealed. Original representative western blots are found in Source Data file.

**Toxicity in primary cultured neurons**. Cortical neurons were obtained from E14 wt mice (C57BL6J). Mice were maintained in accordance to European law and following Hospital Ramón y Cajal's animal guidelines. Cells were obtained using the neuron isolation kit with papain (Thermofisher) and then maintained 7 days in neurobasal medium and treated the last 24 h with 0.2, 2, 10 and 20 μM of CPZ or **3b** or with the same volume of the vehicle DMSO. Three independent experiments counting cells from three different wells per concentration were performed. To analyzed cell death, neurons were fixed and stained with DAPI.

**Parallel artificial membrane permeability assays**. Methodology and data on the permeability in the PAMPA-BBB assay of 10 commercial drugs and compound **3b**, linear correlation between experimental and reported permeability of commercial drugs (see Supplementary Fig. 18, Supplementary Table 5 and Supplementary Methods).

**Protein expression and purification**. See Supplementary Methods.

**Crystallization and diffraction data collection**. Detailed information is provided in Table 1, Supplementary Fig. 17 and Supplementary Methods.

**Fly locomotor activity assay**. Fifteen-day-old males were placed individually in locomotor activity monitor tubes (DAM2, TriKinetics Inc.) The DAM2 system automatically counts the number of beam breaks for flies walking in a horizontal tube over a specific period of time. This setup allowed for characterization of the locomotor and behavior rhythms of *Drosophila*. The tubes contained fly food with

100 µM compound **3b** or same volume of DMSO. Flies were raised at 25 °C in 12-h light/12-h dark. In the first 2 days flies get habituated and the next 2 days the locomotor activity was quantified.

**Synapse number quantification**. The *Drosophila* neuromuscular junction (NMJ) allows the accurate quantitative determination of the in vivo effects of drug application on a single glutamatergic synapse. Each presynaptic motor neuron and postsynaptic muscle fiber can be easily identified and has a stereotypical morphology with minimum inter-individual variability.

We studied the 20-day old male NMJ from the third abdominal hemisegment. Synapses were visualized under confocal microscopy by the nc82 marker (DSHB Hybridoma Bank), which identifies the Bruch pilot protein, a constituent of the presynaptic active zone. Throughout the text, we refer to nc82-positive spots as mature synapses. Neuronal membranes, delimitating motor neuron terminals were revealed by rabbit anti-HRP antibody (Jackson ImmunoResearch). Serial 1-µm confocal images were acquired in a Leica TSC SP5 Confocal Microscope and quantified by Imaris software (Bitplane). Experimental and control genotypes were run in parallel, and quantifications were done blindly.

**Reporting summary**. Further information on research design is available in the Nature Research Reporting Summary linked to this article.

## Data availability

The atomic coordinates and structure factors of the *h*NCS-1/**3b** complex have been deposited in the Protein Data Bank under accession code 6QI4. A reporting summary for this Article is available as Supplementary Information file. The source data underlying Figs. 2, 6 and 9 as well as Supplementary Figs. 1, 2, 9, 10, 15, 18 and Supplementary Table 5 are provided as a Source Data file.

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

## Acknowledgements

R.P.-F., M.J.S.-B. and A.Mansilla thank Prof. J. Elguero (IQM-CSIC), Prof. J. E. Verdasco (UCM) and Dr. L. Infantes (IQFR-CSIC), for helpful discussions on the paper and Prof. A. Ferrús for guidance in the *Drosophila* experiments. M.J.S.-B. thanks Alba Synchrotron for Xaloc Beamline access and support of the staff. This work was supported by different funding agencies: the Spanish Ministry of Economy and Competitiveness (MINECO) with Grants CTQ2015-69643-R (R.P.F), SAF2015-74507-JIN (A. Mansilla) and BIO2017-89523-R (M.J.S.-B.), and European Cooperation in Science and Technology (COST) action CMI304 Emergence and evolution of complex chemical systems (R.P.-F.), and a 2017 Leonardo Grant for Researchers and Cultural Creators from the BBVA Foundation (M.J.S.-B) and a 2018 CaixaImpulse program from La Caixa Foundation (A.Mansilla). M.J.S.-B. was supported by a Ramón y Cajal Contract RYC-2008-03449 from MINECO.

## Author contributions

M.J.S.-B, A.C., D.M., M.E.F.-V., E.S., F.J.C., J.J.-B, R.P.-F. designed the work; A.C.-M., S.B., N.P., R.P.-F performed the chemistry work, J.S., A.C., D.M., M.E.F.-V, E.S. performed NMR studies; M.J.S.-B. supervised protein production (L.M.-G., P.B.-G.) and fluorescence assays (A.C.-M.); P.B.-G. and M.J.S.-B. performed X-ray studies; A.Mansilla performed inmunoprecipitations, toxicity studies and supervised the in vivo studies (A.S.); E.G.-R. and S.M.-S. contributed with the computational studies; J.S., M.J.S.-B, D.M., A.Mansilla contributed drafting the paper, F.J.C., J.J.-B. and A.Martínez revised the manuscript and R.P.-F. wrote the paper.

## Additional information

**Competing interests:** The Spanish National Research Council and the Biomedicine Research Foundation of Ramon y Cajal Hospital have filed the patent applications (P201830933 and EP19382242.6) with the Spanish Patent Office on the use of the compounds described in the paper as synaptic modulators. R.P.-F., A.C.-M., A.Mansilla and M.J.S.-B. are listed as inventors. The remaining authors declare no competing interests.

