## [Peer Review File · Nature Communications]

Reviewers' comments:

Reviewer #1 (Remarks to the Author):

The manuscript describes the use of Dynamic combinatorial chemistry to identify ligands for Neuronal Calcium Sensor-1. A compound, 3b is identified which promotes NCS-1/Ric8a protein-protein interaction, although binding to NCS-1 is relatively weak, with an equilibrium binding constant of about 40 μ M. My principal concern relates to the modelling of a single molecule of 3b into the Ca²⁺ bound hNCS-1 structure. The diffraction data are to a decent resolution (about 1.8 Å) but the density for the bound 3b ligand is weak (Fig 6a,b). This is clear from the very high median B factor for the ligand (97 Å², in the Supplementary Material). It is not easy to tell from the figure, but density for the indole ring looks particularly poor. The contour level for the 2Fo-Fc map in Fig 6a is 0.7 sigma, which is very low- a value of 2 or 3 would be more appropriate. The Polder OMIT map looks better, contoured at 2.5 sigma but, in Phenix, the OMIT map is determined with the ligand bound- it is not clear whether adequate measures have been taken to eliminate phase bias. I suspect these problems arise from the weak binding affinity but I wonder if the authors can justify the conformation of the docked 3b molecule as it is presented- their comparison with the recognition of other ligands would not be reliable if not.

Reviewer #2 (Remarks to the Author):

Insights into Real-Time Chemical Processes in a Calcium Sensor Protein-Directed Dynamic Library

The article represents an interesting addition to the field of protein-templated DCC. This work complements that of Greaney et al.. The novel part of this study is that the authors have shown that DCC can also be applied at low temperatures, i.e. 4 degrees Celsius, which is highly relevant for unstable proteins. They have combined it with extensive NMR studies to obtain real-time information on the intermediates of acylhydrazone exchange. Acylhydrazone chemistry is well studied and known to be a suitable condensation reaction for protein-templated DCC. This study represents the first report and detailed study of the reaction mechanism. An X-ray crystal structure confirms how ligands and target interact. They also introduce p-phenylenediamine as a superior nucleophilic catalyst.

I recommend publication of this paper after the following points have been addressed:

The DCC can be considered as a proof of principle, since the library is really small and no other DCLs are included. What is missing in this article is a table with the relative peak area and in context to this the amplification factors for each molecule in the DCL. This needs to be included. On page 3, fig. 1. the concentration of the hydrazides and aldehyde should be added. In figure 1c. the curve for aniline does not seem to fit too well.

For the protein-protein interaction as mentioned on page 8 in Figure 5. Have these studies been performed in duplicate or triplicate? The deviation gets larger at 10 and 20 on the x-axis. Also, the label for this axis is missing, is it concentration in μ M or mM?

Methods:

- 1) Why is it chosen to work at 3.3 % DMSO, is there a particular reason? For the templated-DCL, is it checked that all binders have come off by filtration of the protein? Please provide some proof for this. Or is the protein also denatured to be sure that the binders are released?
- 2) Did the authors check the maximum concentration of DMSO tolerated by the protein?
- 3) Did the authors check for stability of the protein under the DCC conditions for the time required to reach equilibrium?
- 4) Were the DCC experiments carried out in duplicate/triplicate? For reproducibility, at least duplicates would be required.
- 5) What kind of device is used for the MS part in LC-MS analysis? Please add this information. And can this mass be used to fully characterize the synthesised compounds (like HRMS)? Please use the term protein-directed DCL or target-directed DCL, instead of templated-DCL. Like for example used on page 12.

Page S4, line 68. Add the letter 'l' to catalyst \diamond catalyst.

p. 2, line 80: Greany should be spelled Greaney

p.2, line 77: The authors should cite two more important review articles: P. Freu, R. Hevey and B.

Ernst Chem. Eur. J. 2018 (<https://doi.org/10.1002/chem.201803365>) as well as R. van der Vlag and A. K. H. Hirsch in *Comprehensive Supramolecular Chemistry II*, Elsevier 2017, 5, 487-509.

Reviewer #3 (Remarks to the Author):

The article entitled « Insights into Real-Time Chemical Processes in a Calcium Sensor Protein-Directed Dynamic Library » describe the use of Dynamic Combinatorial chemistry (DCC) and Ligand-based NMR to discover a potent binder of the NCS-1/Ric8a interaction for application in neurological pathologies.

The paper is well written and provide important methodologies for the scientific community working in the discovery of compounds targeting challenging protein-protein interactions. Also, it discloses early data in the research for new treatments of neuro-pathologies. The statistics are well described and the experimental part is well detailed allowing reproducibility by others.

First, the authors describe the DCC experiment. The methodology is inspired by previous reports (ref 19). The experimental originality resides here in the work at low temperature. The technique is well described and the kinetic studies done prior to the selection of the aniline catalyst are very well designed and performed. However, the design of the DCC experiment itself is not sufficiently addressed. Elements that guided the choice of reagents should be detailed. Indeed, only one aldehyde and 5 hydrazides were selected, which is a small number compared to other DCC experiments. The reactivity of the aldehyde, that bears a para-nitro group, may be the reason why it was selected, but the rationale for hydrazides selection is not clear. Is it based on previous compounds identified?

Compound 3b was selected for further studies in the light of NMR studies though LCMS showed that 3e was also a good candidate. Kd values measured by fluorescence showed no clear differences for 3 of the 4 compounds. 3a seems even slightly more affine than 3b. Could the authors provide the Kd value also for 3e and discuss the correlation of the LCMS, NMR and Kd results?

As opposed to chlorpromazine and analogues described in PNAS 2017 and ACS Med Chem Lett 2018 by the authors, the identified compound seems to promote NCS-1/Ric8a complex formation, instead of inhibiting it. Only 3b was tested on the HEK cells for in cellulo proof of concept of the interaction with both partners of the PPI. It would be very important to have the other 3 compounds tested in that model, to see whether the promotion of interaction between NCS-1/Ric8a is series-related or compound-related. Authors mention potential medical applications too for such compound profile. This should be addressed in the introduction when discussing the protein-protein interaction of interest: inhibitors are looked for in certain pathologies while promoters are looked for in others. Maybe a general scheme could be added at this point.

The structures of complexes of hNCS-1/3b are very well detailed and are adequately compared to the structures of hNCS-1 with inhibitors, revealing thus the importance of the interaction of 3b with h10. The last comment in the legend of Fig6, on the superimposition of 3b with one of the two poses of CPZ in hNCS-1 (PDB code: 5g08) is not clear to the reviewer. Could the authors discuss this in the text?

The immune-precipitation assay is an elegant assay for target engagement in a cell model. Nevertheless, compound 3b should be tested in model related to a pathology to validate the concept. The last sentence mentions on-going animal studies. It would be valuable to incorporate these results in the present paper or at least some cellular data in a disease model. 3b was shown to be devoided of cellular toxicity but additional parameters such as LogD, solubility, stability in culture medium; permeability of BHE (BHE PAMPA for example) should be added, as well as one example of pharmacological activity in cells.

The last add-ins regarding a proof of concept in a disease model would definitely strengthen the impact of the paper beyond the elegant methodology. The reviewer thus recommends to accept this manuscript providing major revision.

Minor comments

- The color code for STD signal is misleading as the brown chosen for 70-80% is darker than red > 80%.
- the authors mention the fact that hydrazones can exist either as E or Z isomers. In the experimental part however, compound 3c and 3a were only obtained (synthetically) as the E isomer. 3b was obtained as the E/Z mixture, and was co-crystallized as the E isomer. Could the authors comment on this?
- Please add a figure with the structures of inhibitors identified in previous papers to better present the subject
- Regarding conditions used for the DCC experiment, the rather high percentage of DMSO used (5%)
- Please rewrite ref 16 now bibliographic data is fully available
- In the UF-NMR experiment part, sign needs to be removed before μL (twice).
- “Synthesis of acylhydrazones” should be replaced by “Synthesis of acylhydrazones”
- “Co-Immunoprecipitation” assays should be replaced by “Co-Immunoprecipitation”

We would like to thank you for reviewing our manuscript “Insights into Real-Time Chemical Processes in a Calcium Sensor Protein-Directed Dynamic Library”. Your inputs have been very helpful for improving the manuscript, which has been revised and completed. The modifications have been underlined in the main manuscript.

Point-by-point response to the concerns raised by the reviewers:

Reviewer 1

*The manuscript describes the use of Dynamic combinatorial chemistry to identify ligands for Neuronal Calcium Sensor-1. A compound, **3b** is identified which promotes NCS-1/Ric8a protein-protein interaction, although binding to NCS-1 is relatively weak, with an equilibrium binding constant of about 40mM. My principal concern relates to the modeling of a single molecule of **3b** into the Ca²⁺ bound hNCS-1 structure. The diffraction data are to a decent resolution (about 1.8Å) but the density for the bound **3b** ligand is weak (Fig 6a,b). This is clear from the very high median B factor for the ligand (97Å², in the Supplementary Material). It is not easy to tell from the figure, but density for the indole ring looks particularly poor. The contour level for the 2Fo-Fc map in Fig 6a is 0.7 sigma, which is very low- a value of 2 or 3 would be more appropriate. The Polder OMIT map looks better, contoured at 2.5 sigma but, in Phenix, the OMIT map is determined with the ligand bound- it is not clear whether adequate measures have been taken to eliminate phase bias. I suspect these problems arise from the weak binding affinity but I wonder if the authors can justify the conformation of the docked **3b** molecule as it is presented- their comparison with the recognition of other ligands would not be reliable if not.*

We agree with the reviewer that the maps and high B factors observed for compound **3b** suggest a partial occupancy of the ligand. During refinement, if **3b** occupancy is set to 0.73 and the low-resolution cut-off to 42.35 Å, **3b** median B factor drops to 64.9 Å², a value that now is similar to the surrounding protein. Average B for NCS-1 molecules is 50 Å² and the values for the side chains of the **3b**-coordinating residues are: 44.8 (Trp30), 70.4 (Asp37), 51.6 (Phe48), 57.8 (Tyr52), 88.8 (Phe55), 36.4 (Val68), 42.2 (Phe 72), 42.1 (Phe85), 48.0 (Leu89), 31.9 (Thr92) and 38.5 (Trp103) Å². Also, water molecules interacting with **3b** show similar B-factors: 76.6 (w155), 67.28 (w158), 55.98 (w167), 54.06 (w188), 59.26 (w177). We think that it would be more appropriate to present this structure and structure factors to the Protein Data Bank. The new code, 6QI4, replaces the previous 6H32. From the structural point of view this protein/ligand structure is identical to the previous one. However, and thanks to the reviewer's comments, this structure and resolution cut-off yields to better maps. As suggested, we have performed extra calculations to provide more evidences on the appropriate modeling of compound **3b** bound to NCS-1 (see Results “The crystal structure of hNCS-1 bound to **3b**”, new Fig. 7 and Supp. Info. S17).

Two facts contribute to the partial occupancy of **3b** in the crystal. Firstly, the moderate affinity that the molecule presents for NCS-1 and secondly, the technique used to obtain the crystal structure of the complex: a soaking of native crystals into a **3b**-enriched solution (unfortunately co-crystallization did not work). In the absence of **3b**, native crystals (PDB 1g8i, Bourne et al. *J. Biol. Chem.* **276**, 11949 (2001)) show that the corresponding NCS-1 molecule displays an hydrophobic crevice full of PEG molecules that partially occupy **3b** binding site (see tubular densities in the 2Fo-Fc map, Supp. Fig. b). Therefore, during soaking, **3b** competes with two unspecifically-bound PEGs and moves them away for proper binding (see arrows in Supp. Fig.16b). The **3b** modeled pose fits well with the 2Fo-Fc map of the hNCS-1/**3b** crystal (see zoomed-in view in Supp. Fig.16). Although the reviewer is right that a sigma

level of 0.7 sounds a bit low, I would like to mention that this is not a fixed parameter and depends on the crystal. In this case, neither the native nor the ligand-bound $2F_o-F_c$ maps are noisy at that sigma level suggesting that the map is informative. In fact, to model the structure a sigma level of 0.9-1 was used.

To support the unbiased nature of the protein/ligand structure, we now provide, a F_o-F_c difference map from the first cycles of refinement, before the ligand was built (Supp. Fig.16, orange map). At 2.5σ , two clear blobs of positive density with planar, but not tubular-like shape appear. Those extra densities show no continuity with the PEG molecules, suggesting the presence of a different molecule.

The Polder OMIT map (shown in purple in Supp. Fig.16) is also relevant here. Polder OMIT maps are improved OMIT maps where bulk solvent is excluded in the OMIT region. Therefore in these calculations both the atomic model and the bulk solvent are excluded. In doing so, Polder maps help to enhance weak features in electron density maps and this is particularly useful to demonstrate the presence of a ligand bound to a protein (Liebschner D. *et al. Acta Cryst.* **D73**, 148 (2017)). We have calculated a new Polder map, setting the solvent exclusion radius to 3Å. The sigma level selected for representation is 3.4σ . The density quality reinforces the presence of **3b**. Furthermore, Polder OMIT calculations also include a list of correlation coefficients to assess the confidence of interpreting the map. If CC(1,3) is larger than CC(1,2) and CC(2,3), the density likely corresponds to the atomic features of the Polder OMIT selection. In our case, the map correlations are CC(1,2): 0.7387, CC(1,3): 0.8152 and CC(2,3): 0.6739.

Additionally, we have calculated a Phenix Feature-Enhanced Map (Fig. 7a). These calculations modify $2mF_o - DF_{model} \sigma_A$ -weighted maps. The resulting maps possess a reduced level of noise and model bias and also show enhancement of weak features, often bringing them onto the same scale as the strong features (Afonine, P. V. *et al. Acta Crystallogr. D. Biol. Crystallogr.* **71**, 646-666 (2015)). Our calculated FEM map, shown at 1.4σ , unambiguously supports the conformation modeled and the indole group density is improved with respect to the $2F_o-F_c$ map. Also the map clearly suggests that the nitro group has been well positioned.

There is a lack of density for **3b** hydroxyl group, which is pointing to the solvent. We have tested a **3b** conformation with the -OH group pointing to the protein, which would imply 180° rotation of the 2-hydroxy-3-nitrophenyl ring. We have modelled it into the electron density map and refined it to further analyse the resulting maps (see Supp. Fig.16, right-hand side zoom). The refined conformation, shown in magenta sticks, does not fit well the resulting $2F_o-F_c$ map and its quality is worse compared with that obtained with the final conformation. Also, the positive (green) and negative (red) signal of the F_o-F_c difference map support the final conformation modeled. Additionally, from a chemical point of view it would be energetically unfavourable the positioning of the -OH group towards NCS-1, pointing to V68 and W103 side chains (Fig. 7c).

In addition to the crystallographic calculations, STD-NMR experiments (Fig. 5a) are coincident with the interactions observed in the crystal structure. We now explain in the Results section “The crystal structure of hNCS-1 bound to **3b**” the agreement between both experimental data.

Reviewer 2

I recommend publication of this paper after the following points have been addressed:

-The DCC can be considered as a proof of principle, since the library is really small and no other DCLs are included. What is missing in this article is a table with the relative peak area and in context to this the amplification factors for each molecule in the DCL. This needs to be included.

The selection of the aldehyde and the 5 acylhydrazides was based on previous DCL experiments in which the building blocks reactivity and their concentrations were carefully assessed to ensure the full solubility of the different components. Following the reviewer's suggestion we have additionally included a table with the relative peak area (RPA) and the normalized change of RPA (*Frei, P. et al. Chem. Eur. J.* **23**, 11570 (2017) and *Frei, P. et al. Chem. Eur. J.* **25**, 60 (2019)) to quantify the protein influence in the final outcome (Supp. Info S10). An explanatory paragraph and the corresponding references have been included in the main text.

-On page 3, Fig. 1. the concentration of the hydrazides and aldehyde should be added In figure 1c. The curve for aniline does not seem to fit too well.

The concentrations of the acylhydrazide and the aldehyde have been included in Fig. 2c legend (previous Fig. 1). Thanks to the reviewer's observation the curve for the aniline has been re-plotted including all the points in the adjustment.

-For the protein-protein interaction as mentioned on page 8 in Figure 5. Have these studies been performed in duplicate or triplicate? The deviation gets larger at 10 and 20 on the x-axis. Also, the label for this axis is missing, is it concentration in μM or mM ?

All the experiments have been performed at least three times. A sentence has been included in Fig. 6 legend (previous Fig. 5). In Fig. 6a (Representation of the fluorescence emission of Ca^{2+} loaded *d*NCS1 at increasing concentration of ligand) the unit of axis X has been corrected to μM . Units have also been added to axis X in Fig. 6d. Regarding the deviation observed at 10, 20 μM in Fig. 6d, it must be taken into account that cell toxicity assay in neurons was performed in three independent cultures (from mouse embryos from three different pregnant mothers) where the error bars represent the natural variation of this *in vivo* experiment. Due to the natural heterogeneity of the cultures, seems reasonable, that the error bars tend to be higher when the drugs start having an effect over the cells.

Methods:

1) Why is it chosen to work at 3.3 % DMSO, is there a particular reason? For the templated-DCL, is it checked that all binders have come off by filtration of the protein? Please provide some proof for this. Or is the protein also denatured to be sure that the binders are released?

The percentage 3.3% of DMSO represents accurately the total amount of DMSO in the DCL and DMSO was used for solubility reasons. We have included in Supp. Info. S15 the protein stability studies under our experimental conditions.

The protein was denatured to be sure that the binders were released.

2) Did the authors check the maximum concentration of DMSO tolerated by the protein?

The experiments regarding protein stability are included in Supp. Info. S15. To monitor the tolerance of *d*NCS-1 to DMSO, Tryptophan (Trp) emission fluorescent spectra were recorded at increasing concentrations of DMSO. Trp residues are particularly valuable probes since the indole ring is very

sensitive to its environment. When a protein is unfolding, and due to the progressive exposure of tryptophans to the solvent, the emission maxima (λ_{max}) of the protein spectrum shifts to 350 nm. The presented data (Supp. Info. S15, Fig. 14) indicate that dNCS-1 tolerates up to 20% DMSO since no shift is observed at those concentrations. In fact, at 40% DMSO the protein is mainly folded since a small shift (1nm) is observed. Therefore, the DCC has been performed in conditions where the protein is properly folded. The ability of this protein to bind Ca^{2+} confers stability to the fold and this might be the reason of the tolerance to DMSO.

3) *Did the authors check for stability of the protein under the DCC conditions for the time required to reach equilibrium?*

We have included in Supp. Info (S15, Fig. 15) the NMR of the DCL and protein (100:1) spectra recorded at initial time of mixture and after one and after four days. Zoom on the area where signal of aliphatic side chain residues of the protein appear. Although the base line is very noisy, the profile of the protein signals does not change significantly. The spectra were acquired with 32 scans and processed with 8Hz line broadening for smoothing the noisy base line. Taking into account that the DCL equilibration time is five hours, we are certain that the protein is stable under the DCL experimental conditions.

4) *Were the DCC experiments carried out in duplicate/triplicate? For reproducibility, at least duplicates would be required.*

The DCC experiments were carried out in triplicate. A sentence has been added in Fig.4a legend.

5) *What kind of device is used for the MS part in LC-MS analysis? Please add this information. And can this mass be used to fully characterize the synthesised compounds (like HRMS)?*

A detailed HPLC-MS equipment description plus HPLC and MS methods has been included in Supp. Info. S7. However, HRMS has not been used for the characterization of the synthesized compounds. Other techniques such as NMR and elemental analysis were employed.

Please use the term protein-directed DCL or target-directed DCL, instead of templated-DCL. Like for example used on page 12. Page S4, line 68. Add the letter 'l' to catalyst à catalyist. p. 2, line 80: Greany should be spelled Greaney p.2, line 77: The authors should cite two more important review articles: P. Freu, R. Hevey and B. Ernst Chem. Eur. J. 2018 (<https://doi.org/10.1002/chem.201803365>) as well as R. van der Vlag and A. K. H. Hirsch in Comprehensive Supramolecular Chemistry II, Elsevier 2017, 5, 487-509.

All corrections and the two references mentioned by the reviewer (refs 9 and 10) have been included.

Reviewer 3

First, the authors describe the DCC experiment. The methodology is inspired by previous reports (ref 19). The experimental originality resides here in the work at low temperature. The technique is well described and the kinetic studies done prior to the selection of the aniline catalyst are very well designed

and performed. However, the design of the DCC experiment itself is not sufficiently addressed. Elements that guided the choice of reagents should be detailed. Indeed, only one aldehyde and 5 hydrazides were selected, which is a small number compared to other DCC experiments. The reactivity of the aldehyde, that bears a para-nitro group, may be the reason why it was selected, but the rationale for hydrazides selection is not clear. Is it based on previous compounds identified?

The selection of the aldehyde and the 5 acylhydrazides was based on previous DCL experiments in which the building blocks reactivity and their concentrations were carefully assessed to ensure the full solubility of the different components. The acylhydrazides were mainly aromatic to target the highly hydrophobic NCS-1 crevice but there was not a previous rational design of the library members. A paragraph has been included in the text.

Compound 3b was selected for further studies in the light of NMR studies though LCMS showed that 3e was also a good candidate. Kd values measured by fluorescence showed no clear differences for 3 of the 4 compounds. 3a seems even slightly more affine than 3b. Could the authors provide the Kd value also for 3e and discuss the correlation of the LCMS, NMR and Kd results?

HPLC selected 4 out of 5 E/Z stereoisomers (**3a**, **3b**, **3d**, **3e**). STD-NMR experiments ranked compound **3b** amongst the rest (**3a**, **3d**, **3e**) and fluorescent studies gave a similar binding affinity (micromolar range) except for compound **3e**. The limited solubility of **3e** under the experimental conditions precluded the acquisition of the data and therefore a possible comparison with **3b**. In this revised version we have included co-IP and PAMPA studies supporting **3b** selection as candidate for the *in vivo* studies.

As opposed to chlorpromazine and analogues described in PNAS 2017 and ACS Med Chem Lett 2018 by the authors, the identified compound seems to promote NCS-1/Ric8a complex formation, instead of inhibiting it. Only 3b was tested on the HEK cells for in cellulo proof of concept of the interaction with both partners of the PPI. It would be very important to have the other 3 compounds tested in that model, to see whether the promotion of interaction between NCS-1/Ric8a is series-related or compound-related.

Following the reviewer's comment we have included co-IP studies of compounds **3a**, **3b** and **3d**. The results showed that compounds **3b** and **3d** promote the stabilization of the NCS-1/Ric8a interaction whereas **3a** is an inhibitor similar to CPZ. Therefore, the stabilization effect of **3b** on the NCS-1/Ric8a complex is not believed to be series-related.

Authors mention potential medical applications too for such compound profile. This should be addressed in the introduction when discussing the protein-protein interaction of interest: inhibitors are looked for in certain pathologies while promoters are looked for in others. Maybe a general scheme could be added at this point.

A paragraph and a schematic representation of the regulation mechanism including examples of pathologies associated with an abnormal synapse number (new Fig. 1) were added in the Introduction.

The structures of complexes of hNCS-1/3b are very well detailed and are adequately compared to the structures of hNCS-1 with inhibitors, revealing thus the importance of the interaction of 3b with h10. The last comment in the legend of Fig 6, on the superimposition of 3b with one of the two poses of CPZ

in hNSC-1 (PDB code:5g08) is not clear to the reviewer. Could the authors discuss this in the text?

We have rewritten the sentence for proper understanding. In the same legend (now Fig. 7) we explain that under the crystallization conditions, two different CPZ conformations were modeled, having each of them 0.5 occupancy. One of the conformations binds to the same site as compound **3b** and the rest of inhibitors.

The immune-precipitation assay is an elegant assay for target engagement in a cell model. Nevertheless, compound 3b should be tested in model related to a pathology to validate the concept. The last sentence mentions on-going animal studies. It would be valuable to incorporate these results in the present paper or at least some cellular data in a disease model.

In this revised version we could include the *in vivo* studies in a *Drosophila* model of Alzheimer's disease results from the last two months (see new Fig. 8). These encouraging and positive results show that **3b**-mediated stabilization of the NCS-1/Ric8a complex indeed increases the number of synapses to normal levels exclusively in the presence of a synaptic pathology, which is an essential requirement for any treatment directed to synapses. Furthermore we have evaluated the physiological impact of this synaptic recovery in a fly locomotor activity assay. Remarkably, the locomotor deficit was recovered by **3b** feeding.

***3b** was shown to be devoided of cellular toxicity but additional parameters such as LogD, solubility, stability in culture medium; permeability of BHE (BHE PAMPA for example) should be added, as well as one example of pharmacological activity in cells.*

Following the reviewer's suggestion we have included the PAMPA methodology to predict passive permeability through biological membranes. The *in vitro* permeability (P_e) of **3a**, **3b** and **3d** and ten commercial drugs was then determined. Compounds with $P_e > 4.47 \cdot 10^{-6} \text{ cm} \cdot \text{s}^{-1}$ are able to cross the BBB by passive diffusion. As a result, compound **3b** can be classified as CNS+ with a permeability of $12.9 \pm 0.8 \cdot 10^{-6} \text{ cm} \cdot \text{s}^{-1}$. In contrast, **3a** and **3d** did not show good permeability values (Fig. 6c and Supp. Info. S20). In addition, the *in silico* evaluation of Absorption, Distribution, Metabolism and Excretion (ADME) descriptors such as $\log P_{o/w}$ (pH-independent partition coefficient) and $\log D$ (pH-dependent partition coefficient) were predicted for **3b**, obtaining 2.58 and 2.04 respectively at pH=8 (see Supp. Info. S22). This is in agreement with optimal $\log P_{o/w}$ values (as an indicator of brain-blood partitioning) of 1.5 to 2.5 for drugs targeting CNS. Furthermore, the aqueous solubility ($\log S$) of **3b** was also calculated yielding values of -4.183/-4.675, similar to those obtained for other CNS drugs (see Supp. Info. S22).

The last add-ins regarding a proof of concept in a disease model would definitely strengthen the impact of the paper beyond the elegant methodology. The reviewer thus recommends to accept this manuscript providing major revision.

The proof of concept in a *Drosophila* model of Alzheimer's disease has been included in the manuscript.

Minor comments:

-The color code for STD signal is misleading as the brown chosen for 70-80% is darker than red > 80%.

STD colors have been changed (Fig. 5) following the reviewers' suggestion.

*-The authors mention the fact that hydrazones can exist either as E or Z isomers. In the experimental part however, compound **3c** and **3a** were only obtained (synthetically) as the E isomer. **3b** was obtained as the E/Z mixture, and was co-crystallized as the E isomer. Could the authors comment on this?*

Quantum mechanics (QM) calculations of the geometries for the *E/Z* stereoisomers of **3b** predicted that isomer *E* is preferred; both *in vacuo* and in water (see Sup. Info S22). The calculated pK_a for the acylhydrazone NH (8.0 and 8.5 for *Z* and *E* isomers, see Sup. Info. S22) shows the acidic nature of this NH proton, which strongly suggests that the isomerization from the *Z* to the most stable *E* isomer may easily occur in the reaction medium at pH 8. Interestingly, our crystallographic data and the electron density maps showed that the bound **3b** molecule only displays the *E* geometry, the QM-predicted and most stable isomer (see Supp. Info. S22). Nevertheless, since **3b** is present in solution as a mixture of *Z/E* isomers, the molecular recognition event takes place with a conformational selection process.

-Please add a figure with the structures of inhibitors identified in previous papers to better present the subject.

The chemical structure of the inhibitors has been included in Fig. 7.

-Regarding conditions used for the DCC experiment, the rather high percentage of DMSO used (5%)

Protein stability studies have been included in the Supp. Info. 15. The presented data (S.I. S15, Fig. 14) indicate that dNCS-1 tolerates up to 20% DMSO. Therefore, the protein remains stable under our DCL conditions (< 5% DMSO).

-Please rewrite ref 16 now bibliographic data is fully available. In the UF-NMR experiment part, sign needs to be removed before μL (twice). "Synthesis of acilhydrazones" should be replaced by "Synthesis of acylhydrazones". "Co-Immunoprecipitation" assays should be replaced by "Co-Immunoprecipitation"

All these corrections have been included.

Reviewers' comments:

Reviewer #1 (Remarks to the Author):

The authors have taken my original comments on board and worked hard, using a variety of enhanced difference maps, to justify the positioning of the docked 3b compound. If this were the only evidence for binding to NCS-1 I would be more sceptical but, given the additional data supplied elsewhere in the manuscript, I think there is sufficient evidence to confirm that the modelling of the bound ligand is correct.

Reviewer #3 (Remarks to the Author):

The article entitled « Insights into Real-Time Chemical Processes in a Calcium Sensor Protein-Directed Dynamic Library » describe the use of Dynamic Combinatorial chemistry (DCC) and Ligand-based NMR to discover a potent binder of the NCS-1/Ric8a interaction for application in neurological pathologies.

The revised version of the manuscript has been updated with key experiments and explanations.

Regarding previous comments/questions:

1. The authors clarified somewhat the selection of the 5 hydrazides. Still it is a very small library... Also, the authors discarded 3e because of a lack of response in the fluorescence KD experiment and solubility issues. Still as it had given nice results on the first assay, it seems it shouldn't have been discarded (given also the small library tested here).

2. The authors clarified the introduction and the add-in of the first Figure is a plus.

3. Figure 7 is not clear as the panels are too small. For example, we don't see the PEG in panel c. As well in panel e, the legend mentions helix 10 but it can not be seen as well all the residues that interact with either CPZ should appear.

4. Following the discussion about E/Z mixture of compounds, the authors made quantum analysis for 3b. This study should be done on all compounds for comparison purposes.

5. The authors included an in vivo assay in Drosophila to validate the concept. This is an important improvement. Still, this model is not linked directly to the Protein-Protein interaction that is investigated, as increasing the synapse number can be due to other pathways. It thus needs a target engagement proof. This preclude the use of "proof of concept" phrase. Ideally, it would be key to have a ko-model or if not, the reviewer strongly advise to test 3b in comparison with an inhibitor of the PPI interaction like FD44 and maybe a less active analog as controls to bring some real mechanistic proof of concept that the effect seen is dependent on the PPI.

6. Regarding panel b in Figure 8, it is not clear why the locomotor activity was recorded 4 days but only beam breaks/h for 2 consecutive days are reported. Please explain the difference in time.

7. The authors did some PAMPA measurement, using a brain lipid, and some in silico ADME parameters where gathered. Remove "ADME-Tox studies" as the title of the paragraph as is usually decaded to in vivo studies and as no metabolism and no stability neither on plasma or microsomes was performed. Consider replacing the title of this paragraph by "in vitro permeability, in silico physico-chemical parameters and neuron viability"

8. Phys-chem parameters were calculated and not measured. Only results for 3b are given. As it is in silico, the reviewer recommends to give results for all 5 compounds to document the phys-chem properties in the light of binding. Also for 3e it is necessary as the authors mention solubility problems.

9. p11 line 342: Structure-activity relationships is too strong when there is only 1 compound mentioned! Please reformulate for example with "... to understand binding properties"...

10. The authors conclude now the article that "compound 3b" is a promising candidate for further research in the treatment..." please consider replacing "candidate" by "prototypic probe". Indeed, there are still characterization lacking for this compound like stability, selectivity and specificity to qualify it as a candidate or pharmacological probe.

11. Other minor changes have been taken into consideration.

We thus recommend to publish with minor revision, providing the in vivo experiment contains control compounds like inhibitors of the PPI to ensure a real proof of concept.

Editorial Note: Reviewer #2 was unavailable for the second round of review. Reviewer #4 was therefore invited to assess the authors' response to the comments made by Reviewer #2

Reviewer #4 (Remarks to the Author):

The authors have satisfactorily addressed the main concerns and made the necessary changes to the manuscript. In particular, the proof of product amplification when protein was used by providing relative peak area and detailed calculations based on a reported protocol from Beat Ernst and coworkers. (Chem. Eur. J. 23, 11570 (2017))

However, another question arises while reading the revised manuscript.

In the supporting information on page no S10 and S11, author claim that 'The product distribution was established using independently prepared reference compounds and quantified with each peak area being corrected by the molar extinction coefficient (ϵ) of the corresponding acylhydrazone.' Authors should provide figures of molar extinction coefficient (ϵ) for all acylhydrazones in the supporting information as reported in (Chem. Eur. J. 23, 11570 (2017)).

The manuscript is ready for acceptance after making these minor changes in SI.

We would like to thank you for reviewing our manuscript “Insights into Real-Time Chemical Processes in a Calcium Sensor Protein-Directed Dynamic Library”. Your inputs have been very helpful for improving the manuscript, which has been revised and completed. The modifications have been underlined in the manuscript and added in the Supporting Information.

Point-by-point response to the concerns raised by the reviewers:

Reviewer 1

The authors have taken my original comments on board and worked hard, using a variety of enhanced difference maps, to justify the positioning of the docked 3b compound. If this were the only evidence for binding to NCS-1 I would be more sceptical but, given the additional data supplied elsewhere in the manuscript, I think there is sufficient evidence to confirm that the modelling of the bound ligand is correct.

We are glad that our modelling work has provided convincing insights on this issue and appreciate the reviewer’s positive feedback.

Reviewer 3

The article entitled « Insights into Real-Time Chemical Processes in a Calcium Sensor Protein-Directed Dynamic Library » describe the use of Dynamic Combinatorial chemistry (DCC) and Ligand-based NMR to discover a potent binder of the NCS-1/Ric8a interaction for application in neurological pathologies.

The revised version of the manuscript has been updated with key experiments and explanations. Regarding previous comments/questions:

1. The authors clarified somewhat the selection of the 5 hydrazides. Still it is a very small library... Also, the authors discarded 3e because of a lack of response in the fluorescence KD experiment and solubility issues. Still as it had given nice results on the first assay, it seems it shouldn't have been discarded (given also the small library tested here).

We thank the reviewer for the suggestion. Unfortunately, we had to discard **3e** since it was not possible to have information about its binding properties due to solubility issues. Nevertheless, we will work on the chemical modification of **3e** to improve its solubility.

2. The authors clarified the introduction and the add-in of the first Figure is a plus.

We are glad than the new version of the manuscript is now more clear and appreciate the reviewer’s positive comments.

3. Figure 7 is not clear as the panels are too small. For example, we don't see the PEG in panel c. As well in panel e, the legend mentions helix 10 but it cannot be seen as well all the residues that interact with either CPZ should appear.

A new Fig. 7 has been added to the manuscript following the reviewer's suggestion.

Regarding the raised concerns:

- Figures and labels have been enlarged.
- The complete PEG structure is shown in panel b, however, as explained in the legend, panel c only shows the residues or molecules that are contacting compound **3b** and the PEG molecule the reviewer refers to is not interacting.
- Previous panel e (now panel f) has been redone to show a view similar to that in panel d. Helix H10 is now indicated.
- Given the comment on the suggestion to show the residues recognizing CPZ, we guess that the reviewer would like to see the coordination sphere of an inhibitor vs an stabilizer, since this supports the understanding of the text. In that sense, now include a new panel e showing the coordination sphere of our best inhibitor, **IGS-1.76** (Roca *et al.*, 2018), a molecule that was derived from the structural work carried out with compound **FD44** (Mansilla *et al.*, 2017). CPZ is not a good inhibitor and its structural characterization showed that the molecule is disordered (see the two conformations in panel f). The lack of efficient contacts is the reason for its bad inhibition and therefore is not a good molecule to compare with. This is discussed in the figure legend.

4. *Following the discussion about E/Z mixture of compounds, the authors made quantum analysis for 3b. This study should be done on all compounds for comparison purposes.*

We have performed the calculations for all the compounds and the results are shown in the new Table 5. The energy differences show that, in all cases, the *E* isomer is more stable than the *Z* isomer, both *in vacuum* and in water. The corresponding paragraph has been corrected in the manuscript.

5. *The authors included an in vivo assay in Drosophila to validate the concept. This is an important improvement. Still, this model is not linked directly to the Protein-Protein interaction that is investigated, as increasing the synapse number can be due to other pathways. It thus needs a target engagement proof. This precludes the use of “proof of concept” phrase. Ideally, it would be key to have a ko-model or if not, the reviewer strongly advise to test 3b in comparison with an inhibitor of the PPI interaction like FD44 and maybe a less active analog as controls to bring some real mechanistic proof of concept that the effect seen is dependent on the PPI.*

The final aim of searching for enhancers of the NCS-1/Ric8a interaction is the possibility of restoring synaptic loss with a clear therapeutic application (Summarized in Fig. 1). This aim is particularly relevant since synapse loss is an early symptom of neurodegeneration, which precedes an irreversible neuron death (Selkoe, D. J. 2002 and Herms, J. *et al.* 2016). We have now identified small molecules that enhance the NCS-1/Ric8a interaction, and pursuing this aim, we have tested the best molecule in an A β 42 model, whose synaptic loss has been proven in all experimental systems analyzed to date. We do appreciate the thoughts that the reviewer has provided but we would like to clarify some aspects that may be confusing.

On the reviewer statement: *“increasing synapse number can be due to other pathways”*, although formally possible, it should be realized that the issue has been investigated in *Drosophila* at some length. Following a systematic assay of a number of signaling proteins, two antagonistic pathways were identified, one promoting the formation of synapses, and another repressing synaptogenesis. Both

pathways cross regulate each other providing a scenario in which the actual number of synapses results from the relative balance between both opposing pathways (Mansilla, A. *et al.*, 2018). Beyond the initial signaling to determine the number of synapses, the maintenance and fluctuations of that number, are tightly dependent on the Ca²⁺ dynamics at the neuronal branch endings as it is widely documented in all nervous systems. In that context, the functional mechanism of the Ca²⁺-binding protein NCS-1 and its interaction with Ric8a to control synapse number and probability of release per synapse has been reported (Dason, J. S. *et al.* 2009, Dason, J. S. *et al.* 2012, Romero-Pozuelo, J. *et al.* 2014, Mansilla, A. *et al.* 2017, Mansilla, A. *et al.* 2018).

The use of the proposed KO models have been already documented, mutants, RNAi or dominant negative peptides for NCS-1 or Ric8a, either alone or in combination, were used to characterize the corresponding synaptic phenotypes (Dason, J. S. *et al.* 2009, Dason, J. S. *et al.* 2012, Romero-Pozuelo, J. *et al.* 2014). Building on these previous studies, we set up the assay of small molecules candidates to modify the NCS-1/Ric8a interaction. Consequently, to study the potential changes in the interaction of these two proteins, it is evident that both proteins must be present in the assay system. If a mutant background were used, the interaction would not occur by definition. That is, KO mutants or other loss-of function conditions cannot be used. Regarding the comparison with other compounds, the synaptic effect of the inhibitor **FD44** has also been previously tested in *Drosophila*, but in a model where synapse number was in excess (Mansilla, A. *et al.* 2017), where it was observed the opposite effect to compound **3b**. The suggestion of testing **FD44** in the A β 42 model although possible does not seem logical, since lowering synapse number in a model where this feature is compromised and low seems difficult and most likely non-viable. In this sense, we would like to point out the relevance of testing the molecules in the correct disease model. Finally, and regarding the molecules obtained following the DCC approach, no other compound complies with the binding and PAMPA conditions such as compound **3b**, so its comparison with a less active analog does not seem to be of much interest.

We have now revised the whole text and include more references to make our statements and previous data more clear attending the reviewer's concerns and wishing that the reasoning of our arguments had contributed to clarify the message.

Selkoe, D. J. Alzheimer's disease is a synaptic failure. *Science* **298**, 789-791 (2002).

Herms, J., Dorostkar, M.M. Dendritic spine pathology in neurodegenerative diseases. *Annu. Rev. Pathol.* **11**, 221-250 (2016).

Mansilla, A., Jordán-Alvarez, S., Santana, E., Jarabo, P., Casas-Tintó, S., Ferrús, A. Molecular mechanisms that change synapse number. *J. Neurogenet.* **32**, 155-170 (2018).

Dason, J. S., Romero-Pozuelo, J., Atwood, H. L., Ferrús, A. Multiple roles for frequenin/NCS-1 in synaptic function and development. *Mol. Neurobiol.* **45**, 388-402 (2012).

Romero-Pozuelo, J., Dason, J. S. *et al.* The guanine exchange factor Ric8a binds to the Ca²⁺ sensor NCS-1 to regulate synapse number and neurotransmitter release. *J. Cell Sci.* **127**, 4246-4259 (2014).

Mansilla, A., Chaves-Sanjuan, A., Campillo, N., Semelidoud, O., Martínez-González, L., Infantes, L., González-Rubio, J. M., Gil, C., Conde, S., Skoulakis, E. M. C., Ferrús, A., Martínez, A., Sánchez-

Barrena, M. J. Interference of the complex between NCS-1 and Ric8a with phenothiazines regulates synaptic function and is an approach for fragile X syndrome. *Proc. Natl. Acad. Sci. U S A* 1-10 (2017).

Dason, J.S., Romero-Pozuelo, J., Marin, L., Iyengar, B.G., Klose, M.K., Ferrús, A., and Atwood, H.L. Frequentin/NCS-1 and the Ca²⁺ channel α_1 -subunit co-regulate synaptic transmission and nerve terminal growth. *J. Cell Sci.* **122**, 4109-4121 (2009).

Mansilla, A., Jordán-Alvarez, S., Santana, E., Jarabo, P., Casas-Tintó, S., Ferrús, A. Molecular mechanisms that change synapse number. *J. Neurogenet.* **32**, 155-170 (2018).

6. Regarding panel b in Figure 8, it is not clear why the locomotor activity was recorded 4 days but only beam breaks/h for 2 consecutive days are reported. Please explain the difference in time.

As with other animal models when performing locomotor test, each experiment consists of a habituation phase and a test phase. In our case, flies need a habituation phase of two days. The explanation was in the Materials and Methods section and it has now been added to the Fig. 8 legend for clarification.

7. The authors did some PAMPA measurement, using a brain lipid, and some *in silico* ADME parameters were gathered. Remove “ADME-Tox studies” as the title of the paragraph as it is usually decided to *in vivo* studies and as no metabolism and no stability neither on plasma or microsomes was performed. Consider replacing the title of this paragraph by “*in vitro* permeability, *in silico* physicochemical parameters and neuron viability”

The title has been replaced following the reviewer's comment.

8. Phys-chem parameters were calculated and not measured. Only results for 3b are given. As it is *in silico*, the reviewer recommends to give results for all 5 compounds to document the phys-chem properties in the light of binding. Also for 3e it is necessary as the authors mention solubility problems.

Following the reviewer's suggestion, solubility in water has been calculated for all the compounds and included in the Supp. Info. (see S22).

Physical-chemical properties, which can be directly calculated from the molecular formula or the molecular structure, can be highly accurate or exact. For example, the literature contains a large range of experimentally determined hydrophobicity (logP octanol-water) values and it is possible to generate a highly reliable model of what is essentially a simple physical process. Solubility however is far more complex and depends critically on the data used to train the model (normally experience-based, parametric). Trained datasets span a very wide range of solubilities (around 7 orders of magnitude) and cover a huge range of different chemical structures. However, it is rare to need to consider such a large variation in solubility and structure. Instead it is more normal to consider small variations around a given core. Unfortunately *predicting these subtle variations accurately is extremely hard*, as they are dependent on a huge number of essentially unknown parameters about the nature of the solid state and the ease of liberating molecules from the lattice. This is particularly more complicated when ionizable groups are present in the substrates, such as the phenol group in our small library of ligands. Under our experimental conditions (pH 7.9), the ortho-nitrophenol group (pKa=7.2) is expected to be slightly more than half-dissociated, so both the neutral and anionic species will coexist in significant amounts, both of

them having greatly different solubility. This makes the accurate calculation of the relative solubility of our ligands pH/pKa-dependent and extremely challenging. Slight variations in pH/pKa can translate into huge variations of solubility across very similar ligands.

We tested different solubility predictors for both the neutral and anionic forms of our ligands using different approaches such as fragment-based ChemAxon, 3D-structure-based QikProp in Schrodinger (see table below) and much more sophisticated and time-consuming Free Energy Perturbation molecular dynamics (FEP+) in Desmond-Schrodinger. Despite many attempts, none of these approaches were practical and/or able to reproduce our experimental findings regarding the different solubility observed between **3b** (soluble) and **3e** (insoluble). Most likely, the inability to account for specific interactions (i.e. hydrogen bonds) between the substrate indol NH and water are at the origin of the failure of solubility predictors to reproduce the experimental behavior.

	Method A	Method B (QPlogS)	Method B (CIQPlogS)
3a	-1.6	-3.105	-3.561
3b	-2.4	-4.427	-4.675
3c	-0.3	-2.814	-2.709
3d	-1.7	-3.002	-3.912
3e	-2.0	-3.672	-4.041

Method A: ChemAxon pH-dependent solubility predictor (values roughly estimated at pH 8) from the 2D chemical structure of the ligands

Method B: QikProp solubility predictor in Shroedinger Suite 2018-4. Values calculated for the most abundant protonation state of the lowest-energy conformer calculated in water at pH 8, (QPlogS) or conformer-independent values (CIQPlogS) calculated under the same conditions.

9. *p11 line 342: Structure-activity relationships is too strong when there is only 1 compound mentioned! Please reformulate for example with “... to understand binding properties”...*

Thanks for the reviewer's suggestion. The sentence has been reformulated.

10. *The authors conclude now the article that “compound 3b” is a promising candidate for further research in the treatment...” please consider replacing “candidate” by “prototypic probe”. Indeed, there are still characterization lacking for this compound like stability, selectivity and specificity to qualify it as a candidate or pharmacological probe.*

The phrase has been changed following the reviewer's comment.

11. *Other minor changes have been taken into consideration.*

We thus recommend to publish with minor revision, providing the in vivo experiment contains control compounds like inhibitors of the PPI to ensure a real proof of concept.

Reviewer 4

The authors have satisfactorily addressed the main concerns and made the necessary changes to the

manuscript. In particular, the proof of product amplification when protein was used by providing relative peak area and detailed calculations based on a reported protocol from Beat Ernst and coworkers. (Chem. Eur. J. 23, 11570 (2017)). However, another question arises while reading the revised manuscript. In the supporting information on page no S10 and S11, author claim that 'The product distribution was established using independently prepared reference compounds and quantified with each peak area being corrected by the molar extinction coefficient (ϵ) of the corresponding acylhydrazone.' Authors should provide figures of molar extinction coefficient (ϵ) for all acylhydrazones in the supporting information as reported in (Chem. Eur. J. 23, 11570 (2017)).

The manuscript is ready for acceptance after making these minor changes in SI.

Following the reviewer's suggestion the extinction coefficient has been incorporated to the Supp. Info. (see S26).

REVIEWERS' COMMENTS:

Reviewer #3 (Remarks to the Author):

The authors have satisfactorily addressed the questions in this last version. In particular, more details have been provided for the selection of compounds and the experiment conditions for the in vivo results. Authors added all calculations for E/Z quantum analysis and phys-chem parameters to support the discussion. Also the Figure 7 was clarified and all suggested minor changes have been incorporated. Authors have provided strong answers for the in vivo experiment lacking controls or comparison with other compounds.

In all, we think the manuscript is ready for publication.

We would like to thank you for reviewing our manuscript “Insights into Real-Time Chemical Processes in a Calcium Sensor Protein-Directed Dynamic Library”. Your inputs have been very helpful for improving the manuscript.

Response to the reviewer 3:

Reviewer 3

The authors have satisfactorily addressed the questions in this last version. In particular, more details have been provided for the selection of compounds and the experiment conditions for the in vivo results. Authors added all calculations for E/Z quantum analysis and phys-chem parameters to support the discussion. Also the Figure 7 was clarified and all suggested minor changes have been incorporated. Authors have provided strong answers for the in vivo experiment lacking controls or comparison with other compounds.

In all, we think the manuscript is ready for publication.

We are glad that our calculations for E/Z quantum analysis and phys-chem parameters as well as the detailed provided of the *in vivo* results have afforded convincing insights on these issues and we appreciate the reviewer’s positive comment.